# Gene augmentation prevents retinal degeneration in a CRISPR/Cas9-based mouse model of PRPF31 retinitis pigmentosa

Zhouhuan Xi[1,2,3], Abhishek Vats [1], José-Alain Sahel [1,4,5], Yuanyuan Chen[1] & Leah C. Byrne [1,4,5] ✉

Mutations in *PRPF31* cause autosomal dominant retinitis pigmentosa, an untreatable form of blindness. Gene therapy is a promising treatment for PRPF31-retinitis pigmentosa, however, there are currently no suitable animal models in which to develop AAV-mediated gene augmentation. Here we establish *Prpf31* mutant mouse models using AAV-mediated CRISPR/Cas9 knockout, and characterize the resulting retinal degeneration phenotype. Mouse models with early-onset morphological and functional impairments like those in patients were established, providing new platforms in which to investigate pathogenetic mechanisms and develop therapeutic methods. AAV-mediated *PRPF31* gene augmentation restored the retinal structure and function in a rapidly degenerating mouse model, demonstrating the first in vivo proof-of-concept for AAV-mediated gene therapy to treat PRPF31-retinitis pigmentosa. AAV-CRISPR/Cas9-PRPF31 knockout constructs also mediated efficient *PRPF31* knockout in human and non-human primate retinal explants, laying a foundation for establishing non-human primate models using the method developed here.

Retinitis pigmentosa (RP) is a group of inherited blinding disorders characterized by the degeneration of photoreceptors. Worldwide, RP affects between 1/2000 to 1/3500 people[1,2], and it is a major cause of blindness in younger patients[3–5]. Currently, there is no efficient treatment for RP, making the disease a prime target for the development of novel therapeutics.

The typical onset symptom of RP is night blindness, followed by progressive constriction of the visual field, and culminating in the loss of visual acuity[6–8]. Fundus imaging in patients shows a classic pattern of peripheral pigmentation, known as 'bone-spicule pigmentation', optic disk pallor, and attenuated retinal vessels[9]. Optical coherence tomography (OCT) shows that retinal structure is disrupted, with loss of the inner segment (IS)/outer segment (OS) junction followed by outer nuclear layer (ONL) thinning[10,11]. Electroretinogram (ERG)

recordings in patients often reveal the loss of retinal function, observed as a reduction in a- and b-wave amplitudes, which become non-detectable as the disease advances[12,13]. Histopathologically, RP is characterized by primary degeneration of rods and secondary loss of cones. Alterations in the retinal pigmented epithelium (RPE) and retinal glia activation are also frequently observed[14].

RP is a highly heterogenic group of diseases. Mutations in seven known splicing factors are found to be associated with autosomal dominant RP, including PRPF3[15], PRPF4[16], PRPF6[17], PRPF8[18], PRPF31[19], SNRNP200[20], and RP9[21]. Mutations in PRPF31, pre-mRNA processing factor 31, are the most common among these, accounting for 6–11.1% of autosomal dominant RP[22]. *PRPF31* is a highly conserved gene composed of 14 exons, 1 non-coding and 13 coding, located at chromosome 19q13.4. *PRPF31* encodes a ubiquitous splicing factor, which facilitates

[1]Department of Ophthalmology, University of Pittsburgh, Pittsburgh, PA, USA. [2]Eye Center of Xiangya Hospital, Hunan Key Laboratory of Ophthalmology, Central South University, Changsha, Hunan, China. [3]Department of Ophthalmology, Eye Center, The First Affiliated Hospital of USTC, Division of Life Sciences and Medicine, University of Science and Technology of China, Hefei, Anhui, China. [4]Department of Neurobiology, University of Pittsburgh, Pittsburgh, PA, USA. [5]Department of Bioengineering, University of Pittsburgh, Pittsburgh, PA, USA. ✉e-mail: lbyrne@pitt.edu

the formation and stabilization of the U4/U6-U5 tri-snRNP (small nucleolar ribonucleoprotein) complex and plays essential roles in the mRNA splicing process[23,24]. Homozygous *Prpf31* knockout mice and zebrafish are embryonic lethal, indicating the essentiality of PRPF31[25,26]. Intriguingly, despite ubiquitous expression, in humans, heterozygous mutations in *PRPF31* result in retina-specific disease, for reasons that remain unclear.

PRPF31-RP shows incomplete penetrance as a result of varying levels of PRPF31 expressed by the unaffected allele, which is a phenomenon known as variant haploinsufficiency[1,27]. Expressivity of the *PRPF31* allele follows a continuous distribution in the general population, with the highest expression levels varying by around 5-fold relative to lowest expression levels[28]. A *PRPF31* heterozygous mutation carrier can be asymptomatic if the wild-type (WT) allele produces sufficient PRPF31 for normal retinal function, or can progress to blindness if remaining PRPF31 levels fall beneath a critical threshold. Due to non-penetrance, the prevalence of *PRPF31* mutation carriers may have been underestimated.

Reported mutations in *PRPF31* include frameshift, splice site, missense, and nonsense mutations, as well as replications and large-scale insertions or deletions, most of which are loss-of-function variants[22,29]. Given the wide variety of loss-of-function *PRPF31* variants, and the haploinsufficiency pathogenesis of the disease, gene augmentation represents the most widely applicable therapeutic approach. AAV-mediated gene therapy holds great promise to treat retinal inherited disease, as demonstrated by clinical trials showing the safety and efficacy of gene augmentation to treat LCA2, and subsequent FDA approval of Luxturna, the first clinically approved gene therapy for retinal disease[30]. In a recent study, AAV-mediated *PRPF31* augmentation was shown to improve cell morphology and restore critical functions in iPSC-derived *PRPF31*$^{-/+}$ cells[31]. Yet, to date, there are no suitable animal models in which to develop AAV-mediated *PRPF31* gene therapies. *Prpf31*$^{A216P}$/$^+$ and *Prpf31*$^{-/+}$ mouse models do not develop photoreceptor degeneration up to 18 months of age but show RPE changes that may be more characteristic of age-related macular degeneration than RP[25,32,33]. *Prpf31*$^{+/-}$ zebrafish models also do not show any RP phenotypes[26]. Zebrafish models created through sublethal *Prpf31* morpholino-mediated knockdown[34] and mutant *PRPF31* expression[35] showed disrupted photoreceptor morphology and function. Knockdown of the *PRPF31* homolog in *Drosophila* resulted in anophthalmia or microphthalmia with photoreceptor defects[36]. However, these animals are not useful for preclinical AAV-*PRPF31* testing because AAVs do not infect zebrafish or *Drosophila* efficiently.

The lack of a suitable disease model is a significant obstacle to the development of AAV-*PRPF31* gene therapy. Here, we establish *Prpf31* mutant RP mouse models through the delivery of AAV-CRISPR/Cas9 and characterize the resulting retinal degeneration caused by *Prpf31*-KO. We demonstrate the efficacy of AAV-mediated *PRPF31* augmentation therapy in vivo and lay the foundation for creating NHP models of PRPF31-RP. These experiments provide a new platform for investigating the pathogenesis of PRPF31-RP and developing PRPF31 gene therapies and establish a path to accelerate the translation of AAV-mediated *PRPF31* augmentation therapy to the clinic.

## Results

### Mouse gRNA screening

*Streptococcus pyogenes* Cas9 (SpCas9) and *Staphylococcus aureus* Cas9 (SaCas9) are widely used in vivo genome editing tools[37]. SaCas9 is smaller than SpCas9, allowing for packaging of the Cas9 protein and gRNA into a single AAV vector. In order to create CRISPR/Cas9-based animal models of PRPF31-RP, gRNAs for SpCas9 and SaCas9 were designed to target early coding exons of *Prpf31* in the mouse genome and were screened in vitro for efficiency (Supplementary Fig. 1 and Methods).

For SpCas9, gRNA2 was the most efficient (Supplementary Fig. 1a, b, and Methods). Co-transfection of SpCas9 and gRNA2-GFP plasmids resulted in 92.3 ± 0.58% editing of *PRPF31* in GFP-positive HEK293 cells. For SaCas9, we tested the co-transfection of multiple gRNA constructs to increase efficiency. Co-transfection of SaCas9-gRNAg and SaCas9-gRNAh plasmids outperformed all other gRNA combinations and resulted in the lowest *Prpf31* expression level (39.00 ± 16.09% relative to a GFP-transfected control) in NIH/3T3 cells (Supplementary Fig. 1e, f, and Methods). gRNAg targets mouse *Prpf31* exon 2 and gRNAh targets exon 3.

Having validated a set of efficient gRNAs for use in SaCas9 and SpCas9 constructs, we then administered these constructs using 3 different injection routes to the mouse retina (Fig. 1). We tested subretinal injections (Fig. 1a), in which vector is injected under the retina, resulting in a retinal bubble, or bleb, and high rates of viral transduction in the outer retina. We also tested intravitreal injections (Fig. 1b), a method that is noninvasive to the retina, in which the vector is injected into the fluid filled cavity of the eye, after which vector diffuses across the inner surface of the retina. Lastly, we tested systemic injections (Fig. 1c), in which vector is injected through the facial vein and delivered to the retina via the vasculature, as well as to the entire body.

### Subretinal injection of *Prpf31*-KO vectors results in severe and rapid structural and functional degeneration in photoreceptors and RPE

Degeneration of photoreceptors and RPE in the outer retina is the most prominent pathological feature in RP patients. Therefore, we first aimed to achieve high expression levels of the KO vectors in the outer retina. We injected 7m8-CMV-SaCas9-U6-gRNAg/h (1.5 μl, 1.01E + 12 vg/ml, ratio of gRNAg:gRNAh = 1:1) to the subretinal space of 4 to 5-week-old WT mice, a timepoint at which the retina has fully developed. PBS or a non-targeting (Nt) gRNA mock vector (7m8-CMV-SaCas9-Nt gRNA, 1.5 μl, 4.39 + E12 vg/ml) was injected in control eyes (Fig. 2a). The injection site was made at the temporal retina in a 1.5 μl volume, creating a bleb that covered approximately half of the retina. Eyes were imaged immediately after injection to record the location of the bleb

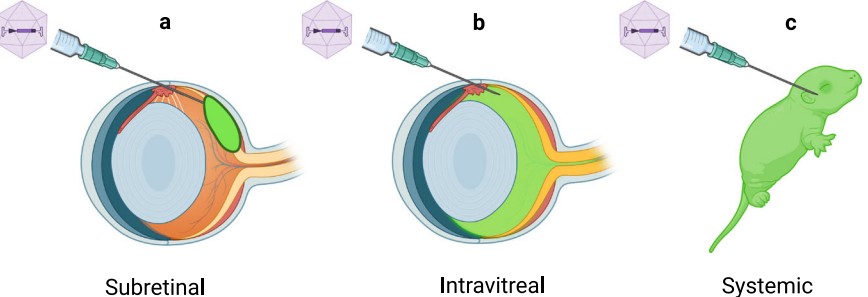

**Fig. 1 | Illustration of AAV administration routes. a** Subretinal injections result in high transduction of the outer retina under the injections site. **b** Intravitreal injections deliver vector across the inner retina. **c** Intravenous injections via the facial vein deliver AAV systemically.

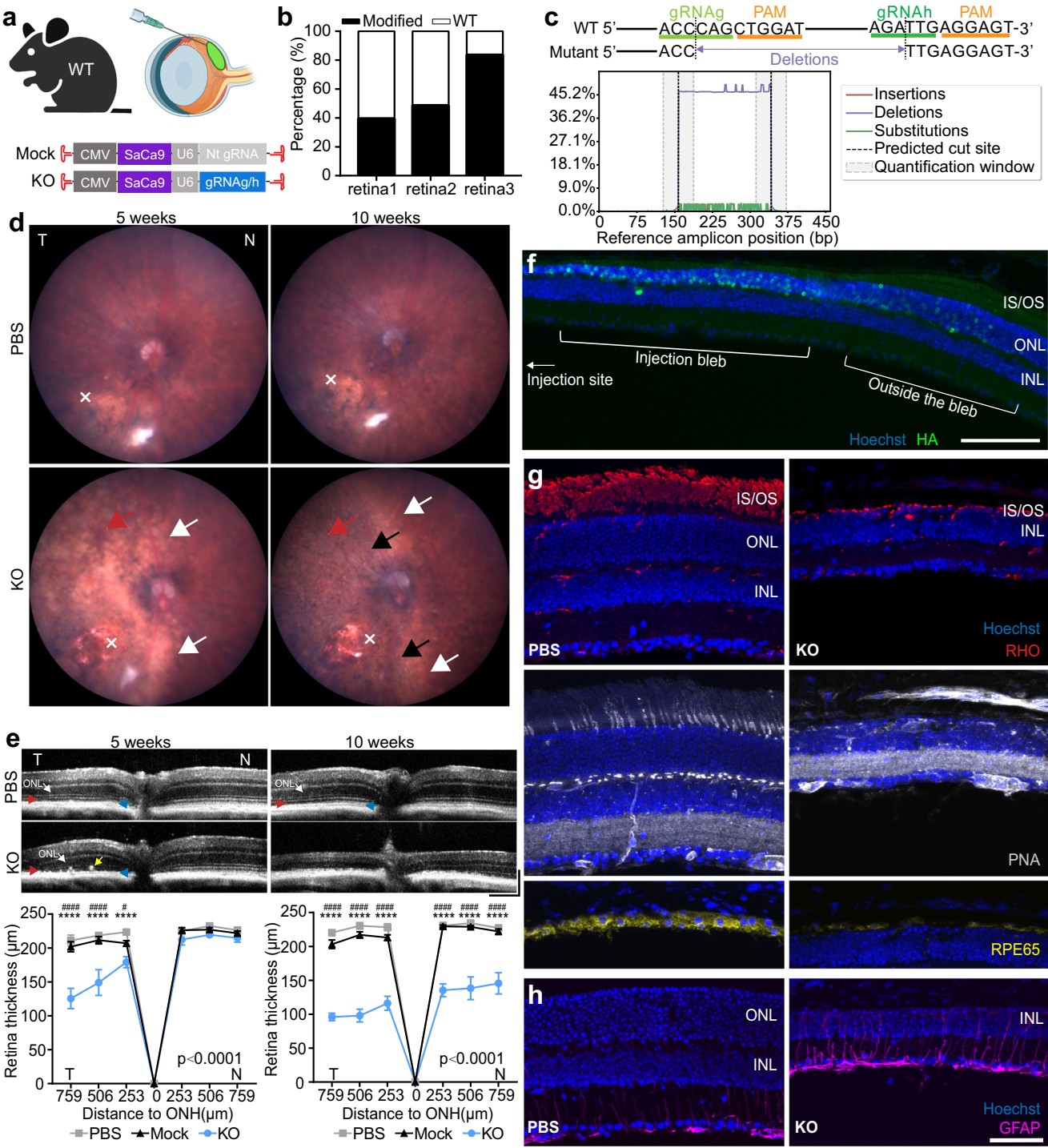

**Fig. 2 | Subretinal injection of two *Prpf31*-KO vectors results in severe structural changes in outer retina. a** Illustration of the experimental approach. AAV 7m8-CMV-SaCas9-U6-gRNAg/h (1.5 μl, 1.01E + 12 vg/ml, gRNAg:gRNAh = 1:1) was subretinally injected to WT mice. PBS or mock vector 7m8-CMV-SaCas9-Nt gRNA (1.5 μl, 4.39 + E12 vg/ml) was injected to control eyes. **b** Deep sequencing of genomic DNA from the injected area of the KO retina 5 weeks (p.i.) revealed 57.29 ± 23.24% editing of *Prpf31* at the gRNAg/h target region (n = 3). **c** Mutations in edited *Prpf31*. The main editing event identified was a -184 bp deletion between the predicted cut sites of the two gRNAs. **d** Fundus images. Injections were made at the temporal retina (×). At 5 weeks (p.i.), temporal retina of KO eyes showed retina pallor (white arrows). At 10 weeks (p.i.), the area of pallor expanded, and punctate black pigmentation (black arrows) and attenuated retinal blood vessels (red arrows) were observed in the injected area. (PBS n = 20, Mock n = 15, KO n = 37) **e** OCT of retina and quantification of total retina thickness (neural retina and RPE)

(n = 6). White arrows, ONL; red arrows, IS/OS junction; blue arrows, RPE layer; yellow arrows, migrating RPE. ONH, optic nerve head; T, temporal; N, nasal. Scale bar vertical, 100 μm; Scale bar horizontal, 200 μm. **f** The expression pattern of HA-tagged Cas9/gRNA vector in the temporal retina of the KO eyes. Scale bar, 100 μm. **g**, **h** Retinal cross sections. KO retinas were lacking ONL and lost markers of rods (RHO in red), cones (PNA in grey) and RPE (RPE65 in yellow). GFAP (magenta) was upregulated in the KO retina. Scale bar, 50 μm. IS/OS = Inner segments/outer segments; ONL = outer nuclear layer; INL = inner nuclear layer. Data are shown as mean ± SEM. For retinal thickness, significance of the overall treatment factor was calculated by two-way ANOVA and shown as a *P* value in each plot. The significance of the treatment factor at each distance from the ONH was calculated by Sidak multiple comparisons test and marked as *P < . 05; **P < . 01; ***P < . 001; ****P < . 0001. Source data are provided as a Source Data file.

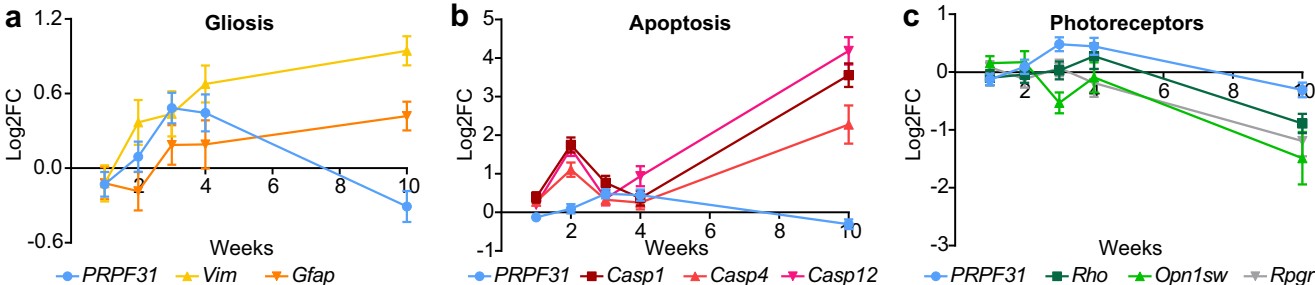

**Fig. 3 | Time course of differential gene expression in KO retina relative to PBS injected retina (n = 3). Expression of *Prpf31* is shown in all panels. a** Marker genes for Müller cell gliosis (*Vim* and *Gfap*). **b** Markers of apoptosis (*Casp1, Casp4,* and *Casp12*). **c** photoreceptor markers (*Rho, Opn1sw* and *Rpgr*). Data are shown as mean ± SEM. For retinal thickness, significance of the overall treatment factor was calculated by two-way ANOVA and shown as a *P* value in each plot. *P* values of gene expression analysis are shown in Supplementary Table 1. Source data are provided as a Source Data file.

(Supplementary Fig. 2). Five weeks post-injection (p.i.), the injected area of the KO retina was dissected out and genomic DNA was subjected to deep sequencing, revealing editing efficiencies of 57.29 ± 23.24% (Fig. 2b). The majority of mutant *Prpf31* contained a ~184 bp deletion between the predicted cut sites (3 bp before the protospacer adjacent motif, PAM) of gRNAg and gRNAh, with the resulting coding sequence out-of-frame (Fig. 2c).

Fundoscopy was used to monitor the condition of retinas in vivo. Fundus imaging showed that 5 weeks (p.i.), the temporal retinas of KO mice were pale (Fig. 2d, white arrows), which overlapped with the location of the injection bleb (Supplementary Fig. 2). These pale patches of retina were not observed in PBS and mock-injected groups, confirming that retinal pallor was not a result of the injection procedure. The area of retinal pallor was expanded at 10 weeks (p.i.), and punctate black pigmentation formed, similar to the characteristic pigmentation observed in PRPF31-RP patients (Fig. 2d, black arrows). Attenuation of the retinal blood vessels, another funduscopic hallmark of RP, was also found in the injected area (Fig. 2d, red arrows).

OCT imaging showed significantly thinned temporal retina in KO eyes 5 weeks (p.i.), with loss of IS/OS, and reduction of ONL thickness. Additionally, the border between OS and the RPE layer became unclear, and migration of the RPE towards the inner retina was observed. Ten weeks (p.i.), dramatic retinal thinning with a complete loss of IS, OS and ONL was observed at the temporal retina. The RPE layer could not be identified. The nasal side (away from the injection site) of the retina also showed the disappearance of IS, OS, and ONL thinning at 10 weeks (p.i.), indicating that the area of retinal degeneration further expanded (Fig. 2e).

To further demonstrate structural changes in the KO eye at the histological level, we labelled HA-tagged Cas9 using anti-HA tag antibody. The expression pattern of Cas9 in the KO retina was the strongest in the outer retina under the injection bleb and decreased towards the edge of the bleb. Decreased anti-HA labeling, along with ONL thinning, was observed towards the center of the injection site due to the death of *Prpf31*-KO cells (Fig. 2f). The KO eyes showed notably reduced labeling of rhodopsin (RHO), peanut agglutinin lectin (PNA), and RPE65, indicating the loss of rods, cones, and RPE, corresponding to observations from OCT imaging (Fig. 2g). GFAP, a marker of Müller cell activation, was increased in KO retinas, indicating gliosis, a phenomenon often seen in degenerating retina (Fig. 2h).

To better understand the molecular mechanisms underlying retinal degeneration caused by *Prpf31* KO, we quantified the mRNA expression of marker genes of temporal retina from KO and PBS-injected eyes 1, 2, 3, 4 weeks and 10 weeks (p.i.) (Fig. 3). Compared to PBS-injected eyes, expression of *Prpf31* in KO eyes initially increased over the course of 3 weeks (p.i.), likely as a compensatory mechanism

for the loss of PRPF31, which was also reported by Yin et al. in a zebrafish model[35], and then decreased significantly from 4 to 10 weeks (p.i.) (Fig. 3a–c). Markers of Müller cell activation, *Vim* and *Gfap*, steadily increased over the 10-week time period (Fig. 3a). Markers of apoptosis, *Casp1, Casp4,* and *Casp12*, initially increased, decreased briefly as *Prpf31* expression transiently increased, and then increased dramatically from 4 to 10 weeks p.i (Fig. 3b). Expression of photoreceptor markers *Rho, Opns1w,* and *Rpgr* decreased slightly from 1–3 weeks (p.i.), and then decreased precipitously from 4 to 10 weeks (p.i.) as *Prpf31* expression decreased (Fig. 3c).

ERG recording was used to evaluate the function of the retina. Scotopic and photopic a-, b- and c-waves were recorded, which reflect the response of photoreceptors, bipolar cells, and RPE, respectively. ERG recordings from eyes 5 weeks (p.i.) showed significantly reduced scotopic and photopic a- and b-wave responses in KO eyes across a wide range of stimulus intensities, indicating photoreceptor dysfunction (Fig. 4a–c). The c-wave amplitudes recorded from KO eyes were dramatically decreased, with prolonged implicit times compared to PBS and mock-injected groups, indicating compromised RPE function (Fig. 4d, e). At 10 weeks (p.i.), ERG responses of KO eyes were further impaired, showing almost no detectable a-, b-, or c-waves, indicating a dramatic loss of photoreceptor and RPE function (Fig. 4a–e).

Together, these data show that subretinal delivery of *Prpf31*-KO vectors results in substantial structural and functional impairment in the outer retina and leads to photoreceptor and RPE degeneration consistent with the typical clinical features of PRPF31-RP patients.

In order to test a single-vector system, we also subretinally injected 7m8-U6-gRNA2-CAG-GFP (0.6 μl, 2.13E + 12 vg/ml) in 4 to 5-week-old SpCas9-expressing mice (H11^Cas9 CRISPR/Cas9 knock-in mice, Jackson labs). GFP was co-expressed with gRNA to allow for in life monitoring of transgene expression. A control vector, 7m8-CAG-GFP (0.6 μl, 2.46E + 12 vg/ml), was injected in the contralateral eye (Supplementary Fig. 3a). At 5 weeks (p.i.), deep sequencing revealed a 14.92 ± 4.10% editing of *Prpf31* DNA from whole retinas of KO eyes (Supplementary Fig. 3b). Over 50% of editing events resulted in the insertion of an adenosine at the predicted cut site of the gRNA2 target sequence, causing a frameshift mutation and generation of a premature stop codon (Supplementary Fig. 3c).

The results of *Prpf31* KO in SpCas9-expressing mice were largely equivalent to those obtained using a 2-vector approach in WT mice. Structural changes, including significant reduction of retinal thickness, loss of photoreceptor and RPE markers, and impairment of retinal function, reflected by defective a-, b-, c- wave, were observed within 10 weeks (p.i.) (Supplementary Fig. 3d–h). Together, these results further validate those obtained using subretinal injections in WT mice and provide a single-vector system in which the effects of PRPF31 mutation can be investigated.

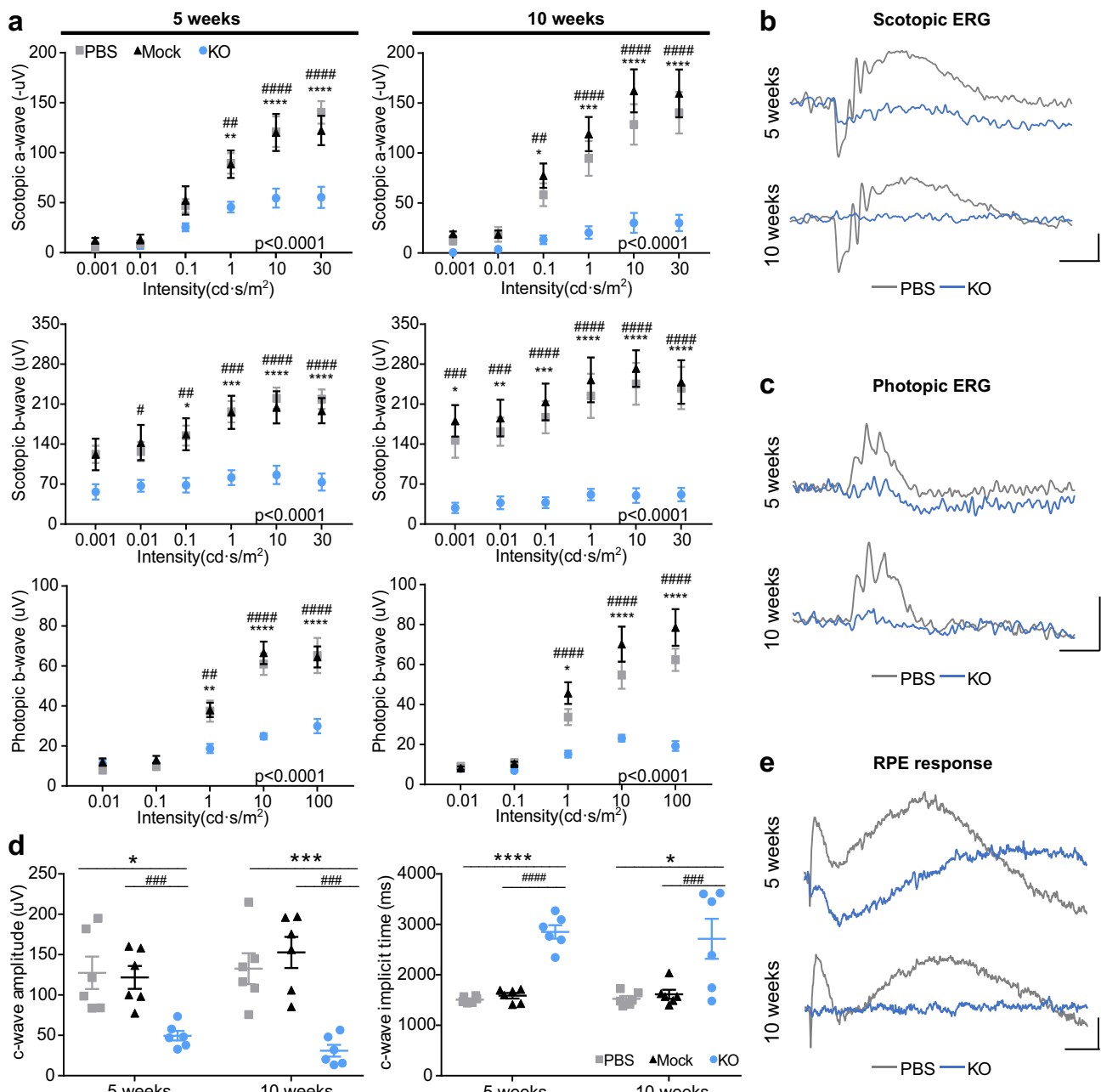

**Fig. 4 | Subretinal injection of two *Prpf31*-KO vectors results in severe functional changes in outer retina. a** Quantification of ERG scotopic and photopic a- and b-wave amplitudes ($n = 6$). **b** Representative scotopic and **c** representative photopic ERG recordings stimulated by a 10 cd·s/m² flash. Scale bar vertical, 50 μV; Scale bar horizontal, 50 ms. **d** Quantification of dark-adapted ERG c-waves ($n = 6$). **e** Representative ERG c-wave recordings. Scale bar vertical, 50 μV; Scale bar horizontal, 500 ms. ERGs recorded 5 weeks (p.i.), showed significantly reduced scotopic and photopic a-wave and b-wave responses in KO eyes across a wide range of stimulus intensities. ERG c-wave responses from KO eyes were dramatically decreased and had prolonged implicit times compared with PBS and mock-injected eyes. At 10 weeks (p.i.), ERG responses of KO eyes were further impaired, showing almost no detectable a-, b-, or c-waves, indicating a dramatic loss of photoreceptor and RPE function. Data were shown as mean ± SEM. The significance of the overall treatment factor was calculated by two-way ANOVA and shown as *P* values in the plots. The significance of the treatment factor at each flash intensity was calculated by Sidak multiple comparisons test and marked as \**P* < .05; \*\**P* < .01; \*\*\**P* < .001; \*\*\*\**P* < .0001. "\*" shows significance between KO and PBS groups, and "#" shows significance between KO and mock groups. Data from ERG c-waves were analyzed using a two-tailed unpaired t-test. Source data are provided as a Source Data file.

## Intravitreal injection of Prpf31-KO vectors results in structural and functional changes primarily in the inner retina

Intravitreal injections are noninvasive to the retina and deliver AAV particles to the vitreous cavity of the eye, resulting in diffusion of vector across the expanse of the retina. With 2nd generation AAV capsids such as 7m8, intravitreally injected AAV particles first encounter cells in the inner retina, but can bypass structural barriers to transfect all retina layers[38]. 7m8-U6-gRNA2-CAG-GFP (2 μl, 2.13E + 12 vg/ml) was delivered intravitreally to SpCas9-expressing mice aged 4 to 5 weeks. AAV 7m8-CAG-GFP (2 μl, 2.46E + 12 vg/ml) was delivered to the contralateral eye as a control (Fig. 5a). Deep sequencing of genomic DNA from whole retinas confirmed the editing of *Prpf31* (3.73 ± 0.19%) 5 weeks (p.i.) (Fig. 5b). The addition of a single adenosine at the predicted cut site of gRNA2 target accounted for the majority of the mutant form of *Prpf31* (Fig. 5c).

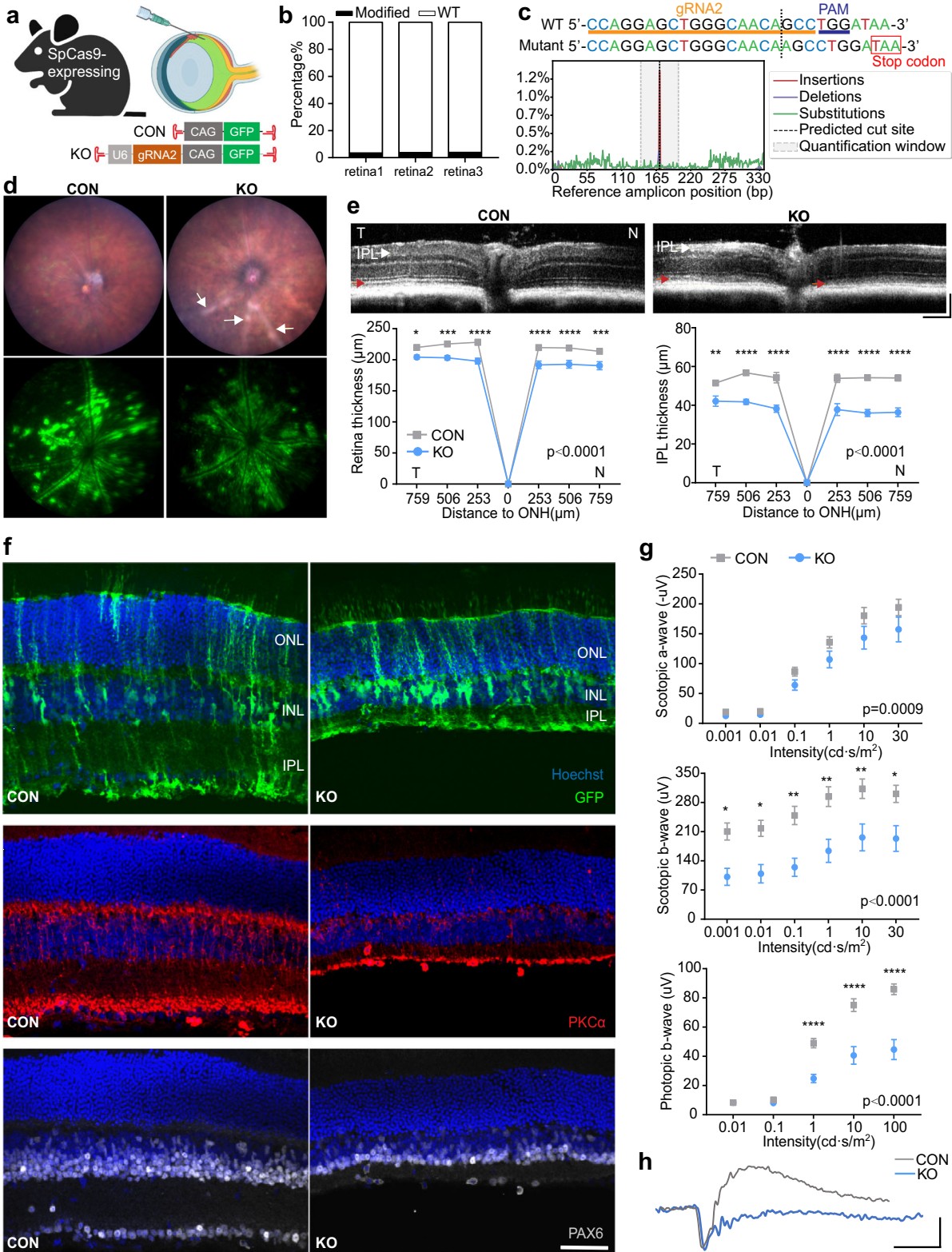

Five weeks (p.i.), expression of GFP was observed across the expanse of the retina (Supplementary Fig. 4a). Quantification of total retinal thickness from OCT imaging showed no statistically significant difference between control and KO groups while the thickness of the inner plexiform layer (IPL) was slightly but significantly reduced (Supplementary Fig. 4b). Fifteen weeks (p.i.), retina pallor was observed alongside the blood vessels in KO eyes (Fig. 5d, arrows and

Supplementary Fig. 5), which overlapped with areas showing the highest GFP expression, suggesting the presence of retinal lesions occurring in the areas with highest gRNA expression. Also, at this later time point, a significant reduction in total retinal thickness was observed in KO eyes, with the greatest loss observed in the IPL. A disturbance of IS/OS junction was also noted in some KO eyes (Fig. 5d, red arrows). Imaging from retinal cross-sections showed that GFP was

**Fig. 5 | Intravitreal injection of *Prpf31*-KO vectors results in structural and functional changes primarily in the inner retina. a** Illustration of experimental approach. KO vector 7m8-U6-gRNA2-CAG-GFP (2 µl, 2.13E + 12 vg/ml) was delivered to SpCas9-expressing mice via intravitreal injection. 7m8-CAG-GFP (2 µl, 2.46E + 12 vg/ml) was delivered to the contralateral eye as control. **b** Deep sequencing of genomic DNA from whole retina of the KO eyes 5 weeks (p.i.) revealed 3.73 ± 0.19% editing of *Prpf31* at the gRNA2 target region (*n* = 3). **c** Mutations found in modified *Prpf31*. The main editing event identified was the insertion of an adenosine at the predicted cut site (3 bp before PAM). **d-h** Retinal conditions at 15 weeks (p.i.) **d** Fundus images. Retina pallor was observed alongside blood vessels in KO eyes (arrows), suggesting retinal lesion (CON *n* = 19, KO *n* = 19). **e** OCT of retinas and quantification of total retinal thickness (neural retina and RPE) and IPL thickness (*n* = 8). Red arrows, IS/OS junction. ONH, optic nerve head; T, temporal; N, nasal. Scale bar vertical, 100 µm; Scale bar horizontal, 200 µm. **f** Retinal cross sections. GFP (green) was expressed in all retinal layers with the strongest expression in the inner retina and Müller cells. KO eyes had significant thinning of the IPL and INL, loss of the bipolar cell marker (PKCα in red) with shortened bipolar processes, as well as loss of the ganglion cell/amacrine cell marker (PAX6 in grey) compared to control eyes. Scale bar, 50 µm. **g** Quantification of ERG scotopic and photopic a- and b-waves (*n* = 9). The photopic and scotopic b-wave amplitudes in KO eyes decreased dramatically, while scotopic a-waves were moderately reduced. **h** Representative scotopic ERG recordings stimulated by a 10 cd·s/m² flash. Scale bar vertical, 50 µV; Scale bar horizontal, 50 ms. Data were shown as mean ± SEM. Significance of the overall treatment factor was analyzed using two-way ANOVA and stated as a P value in the plots. Significance of the treatment factor at each distance from ONH (for OCT) or each flash intensity (for ERG) was calculated by Sidak multiple comparisons test and marked as *$P < .05$; **$P < .01$; ***$P < .001$; ****$P < .0001$. ONH = optic nerve head. ONL = outer nuclear layer; INL = inner nuclear layer; IPL = inner plexiform layer. Source data are provided as a Source Data file.

expressed in all retinal layers, with the strongest expression in the inner retina and Müller cells at 15 weeks (p.i.) KO eyes were marked by notable IPL and INL thinning, in agreement with OCT imaging (Fig. 5e). KO eyes showed a loss of the bipolar cell marker PKCα, shortened bipolar processes, as well as loss of the ganglion cell/amacrine cell marker PAX6 compared to control eyes (Fig. 5f).

ERG recording showed functional impairment of the KO retina. Five weeks (p.i.), scotopic a- and b-wave amplitudes of KO eyes were slightly reduced compared to control eyes (Supplementary Fig. 4c–e). Fifteen weeks (p.i.), b-wave amplitudes of the KO group decreased dramatically, while a-waves were moderately reduced (Fig. 5g, h), indicating the KO eyes had greater dysfunction in bipolar cells than in photoreceptors. While scotopic a-waves were reduced in KO eyes, no obvious reduction in rod and cone cell markers were observed between the two groups (Supplementary Fig. 6). Together, these data show that intravitreal delivery of *Prpf31*-KO vectors caused structural and functional impairment primarily in the inner retina of SpCas9-expressing mice.

## Systemic injection of Prpf31-KO vectors results in abnormal development and high early mortality rate

Systemic delivery is noninvasive to the eye, and AAV92YF has been shown to achieve efficient transgene expression in the retina following systemic injection in neonates[39]. We, therefore, delivered SaCas9 systemically using AAV92YF. AAV92YF-CMV-SaCas9-U6-gRNAg and AAV92YF-CMV-SaCas9-U6-gRNAh were mixed 1:1, and a 50 µl total volume at a titer of 1.35E + 13 vg/ml was delivered to P0 neonatal WT mice via facial vein injection (Fig. 6a). An equal volume of PBS or mock vector AAV92YF-CMV-SaCas9-Nt gRNA at a titer of 3.82E + 13 vg/ml was injected to litter mates as control. Systemic KO of *Prpf31* resulted in pups with low body weight and high early mortality rate within 4 weeks (p.i.) (Fig. 6b–d). Evaluation of retinal structure and function by OCT and ERG was not possible because KO mice were too weak to survive anesthesia. *Prpf31* editing was confirmed at the DNA and protein level. Liver tissue, which is efficiently targeted by systemic injections of AAV92YF, was collected. Genomic DNA was extracted and gRNAg/h target sites were PCR amplified. *Prpf31* DNA from the KO group was truncated, at a length corresponding to the distance between gRNAg and gRNAh (184 bp) (Fig. 6e). Western blotting also showed significantly decreased PRPF31 protein in livers from the KO group (Fig. 6f–g). These results suggest that *Prpf31* is essential for postnatal development and survival.

## Gene augmentation preserves histological structure and visual function in Prpf31 knockout retinas

Each of the 3 routes of injection tested – subretinal, intravitreal and systemic – resulted in genome editing in vivo. The severity of the disease and outcomes of gene editing were largely influenced by the injection route. Each of these models may be useful for the study of

PRPF31. However, of the models tested, subretinal injections most closely resembled the pathophysiology of *PRPF31*-RP, with dramatic loss of photoreceptors and RPE, similar to the pathological changes observed in patients. This rapidly available model enables the testing of AAV-mediated gene augmentation for PRPF31 RP in vivo for the first time. We, therefore, tested a gene augmentation vector encoding human PRPF31 (*hPRPF31*) driven by a ubiquitous CAG promoter. *hPRPF31* is not targeted by gRNAg/h due to a mismatch of the gRNA target sequence or the PAM sequence in the human genome. For each mouse, one eye was co-injected with KO vector 7m8-CMV-SaCas9-U6-gRNAg/h (0.75 µl, 1.01E + 12 vg/ml) and rescue vector 7m8-CAG-hPRPF31 (0.75 µl, 1.05E + 12 vg/ml). Vectors were mixed in a ratio of 7m8-CMV-SaCas9-U6-gRNAg: 7m8-CMV-SaCas9-U6-gRNAh: 7m8-CAG-hPRPF31 = 0.5:0.5:1. The contralateral control eye was injected with KO vector (0.75 µl) and PBS (0.75 µl) (Fig. 7a). The total volume of each subretinal injection was 1.5 µl/eye, and the injection sites were located at the temporal retina. The expression of h*PRPF31* mRNA in the KO-Rescue eyes was confirmed at 10 weeks (p.i.) by qPCR using a pair of primers that bind to h*PRPF31* specifically but not mouse *Prpf31* (*mPrpf31*) (Supplementary Fig. 7). Deep sequencing of genomic DNA collected from whole retina at 10 weeks (p.i.) revealed that 38.02 ± 12.66% of *mPrpf31* was modified in KO-Rescue eyes, while 24.08 ± 3.37% *mPrpf31* was modified in the KO-PBS eyes (Fig. 7b). Following injection with the same dosage of *Prpf31* KO vectors, KO-Rescue eyes had a higher percent of edited *mPrpf31* compared to KO-PBS eyes, indicating that *mPrpf31*-KO cells were preserved in KO-Rescue retinas due to *hPRPF31* augmentation. PRPF31 protein levels, quantified using automated Western blotting (Jess system, ProteinSimple), were reduced in KO-eyes, and were restored to normal levels in KO-Rescue eyes (Supplementary Fig. 8).

Fundoscopy in KO-rescue eyes revealed healthier retinas with reduced retinal pallor and pigmentation than the KO-PBS eyes (Fig. 7c and Supplementary Fig. 9). As previously observed, KO-PBS eyes were marked by temporal retinal pallor at 5 weeks (p.i.) At 7.5 weeks (p.i.), pigmentation formed at the area with retinal pallor and attenuation of blood vessels can be observed. And, 10 weeks (p.i.), the area of retinal pallor further expanded, with increased pigmentation. In contrast, no prominent area of retina pallor was observed in KO-Rescue eyes, and pigmentation was found only in a limited area surrounding the injection site, likely as a result of damage caused by subretinal injection.

OCT scanning showed maintained retinal thickness and preserved normal retinal structure in KO-Rescue eyes (Fig. 7d). In KO-PBS eyes, retinal thinning at the far temporal side (closest to the injection site) was apparent 5 weeks (p.i.), and at 10 weeks (p.i.), the thickness of temporal retina was dramatically decreased. In the affected area, IS and OS disappeared, the ONL was significantly thinned, and the border between RPE and OS was disturbed. In contrast, KO-Rescue retinas were significantly thicker, with only a minor reduction in retinal thickness from 5 weeks to 10 weeks (p.i.). Although in some individual

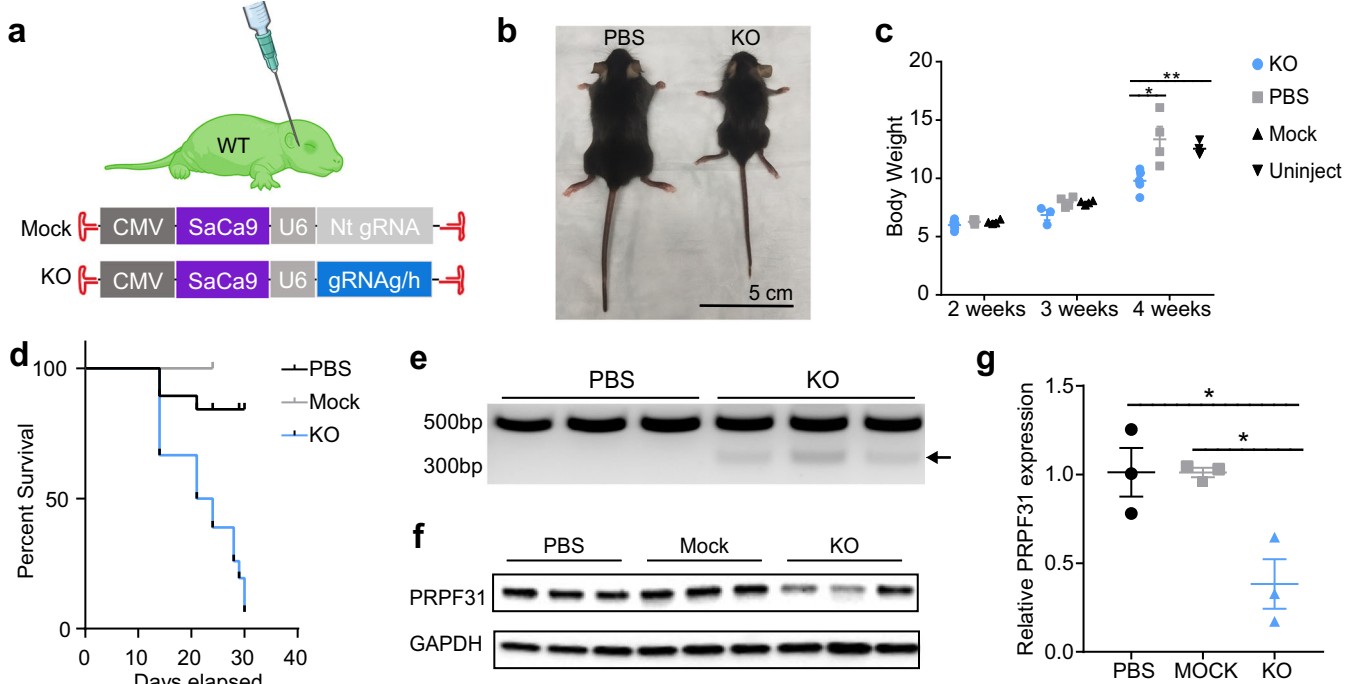

**Fig. 6 | Systemic injection of Prpf31-KO vectors to P0 mice. a** Illustration of the systemic injection approach. AAV92YF-CMV-SaCas9-U6-gRNAg/h (50 μl, 1.35E + 13 vg/ml, gRNAg:gRNAh = 1:1) was delivered to P0 WT mice via facial vein injection. An equal volume of PBS or mock Cas9 vector AAV92YF-CMV-SaCas9-Nt gRNA (3.82E + 13 vg/ml) was injected to litter mates as control. **b** Images of 2 male mice taken 4 weeks (p.i.) Compared with PBS-injected control litter mate, KO pups developed abnormally. **c** Body weights of PBS, Mock and KO vectors injected and uninjected pups 2–4 weeks (p.i.) The KO group had significantly lower body weights (*n* = 3–6). Males and females were evenly distributed in each group. (2 weeks: PBS *n* = 5, Mock *n* = 4, KO *n* = 6; 3 weeks: PBS *n* = 4, Mock *n* = 4, KO *n* = 3; 4 weeks: PBS *n* = 4, Mock *n* = 4, KO *n* = 5). **d** Survival plot of PBS, Mock and KO-injected pups. KO pups had a high early mortality rate (PBS *n* = 16, Mock *n* = 4, KO *n* = 18). The first check point was at P14 to avoid disturbing dams and pups. **e** PCR amplification of gRNA targeted region showed truncated *Prpf31* (black arrow) in DNA collected from the KO group. **f, g** Western blots of liver samples collected from PBS, Mock and KO vectors injected pups. Bar plots of relative PRPF31 band intensity normalized by GAPDH show a significant reduction of PRPF31 protein in the KO group (*n* = 3). Data were analyzed by unpaired two-tailed t-test and are shown as mean ± SEM. **P* < .05; ***P* < .01. Source data are provided as a Source Data file.

KO-Rescue eyes, the border between OS/IS of temporal retina became less defined 5 weeks (p.i.), OS, IS, ONL, and RPE layers were preserved 10 weeks (p.i.). Heatmaps of retinal thickness generated from OCT volume scans of KO-PBS eyes illustrated significant retinal thinning at the temporal retina, while KO-Rescue eyes had improved thickness across the retina with damage limited to the injection site, corresponding to the results from fundoscopy and OCT B-scans (Fig. 7e). Immunolabeling of the HA-tagged SaCas9, photoreceptor and RPE maker genes confirmed the expression of KO vectors, preserved structure of the ONL and OS/IS and improved retinal morphology including survival of rods, cones, and RPE in KO-Rescue eyes (Fig. 7f).

Differential gene expression analysis was performed on mRNA from whole retina from KO eyes, KO-Rescue eyes, mock-injected eyes 10 weeks (p.i.), and WT uninjected litter mates (Fig. 7g). KO-Rescue eyes had higher expression levels of total *PRPF31* (*hPRPF31* and *mPrpf31*), lower expression levels of the cell death marker *Casp12* and the glial cell activation markers *Vim and Gfap*, as well as higher expression levels of photoreceptor-specific gene markers (*Rho* for rods, *Opn1sw* and *Pde6c* for cones), compared to KO eyes, with overall gene expression levels similar to WT uninjected eyes. Collectively, differential gene expression analysis showed that *hPRPF31* gene augmentation reduced cell death and gliosis, while preserving photoreceptor survival.

ERG recordings showed that KO-rescue eyes had improved rod and cone responses compared to KO-PBS eyes at all time points measured (Fig. 8a, c). KO-rescue eyes showed higher dark-adapted a-, and b- wave amplitudes than the KO-PBS eyes 5 weeks (p.i.) At 7.5 weeks (p.i.), a- and b-wave amplitudes of KO-rescue remained similar to amplitudes measured at 5 weeks (p.i.) while KO-PBS eyes showed a severe reduction of a- and b- wave amplitudes. At 10 weeks (p.i.), a-, and b- wave amplitudes of KO-rescue eyes remained significantly higher than those of KO-PBS eyes. The improvement of c-waves in the rescue group was also noted. From 5 to 10 weeks, the KO-rescue eyes had significantly higher c-wave amplitudes and shorter implicit times compared to KO-PBS eyes (Fig. 8b, d).

## KO of PRPF31 in primate retinal explants demonstrates the feasibility of establishing a NHP PRPF31-RP model

Mice are a leading preclinical model to test retinal gene therapies, due to ease of use, availability, and affordability. However, rodent eyes differ from human eyes in several key aspects, particularly, the lack of a fovea. NHPs are the model with eyes most similar to humans, and it is therefore of substantial interest to develop NHP models for preclinical development of gene therapies.

There is no available SaCas9 gRNA that targets early exons of both mouse and human/rhesus macaque PRPF31, due to cross-species sequence differences. We, therefore, designed gRNA constructs targeting human and rhesus macaque *PRPF31*. Co-transfection of SaCas9-gRNAa/c (44.33 ± 6.66% *PRPF31* expression relative to control) and SaCas9-gRNAa/d (51.67 ± 7.57% *PRPF31* expression relative to control) showed the highest knockout efficiency among the combinations tested. The combination of SaCas9-gRNAa/b also resulted in considerably reduced *PRPF31* expression (56.67 ± 9.03% *PRPF31* expression relative to control) (Supplementary Fig. 1c, d, and Methods).

We then packaged and tested the SaCas9-gRNAa/b/c/d vectors in rhesus and human ex vivo cultured retina, to establish proof-of-concept and demonstrate the feasibility of establishing a NHP model of PRPF31-RP using this system. Five-mm round retinal pieces were

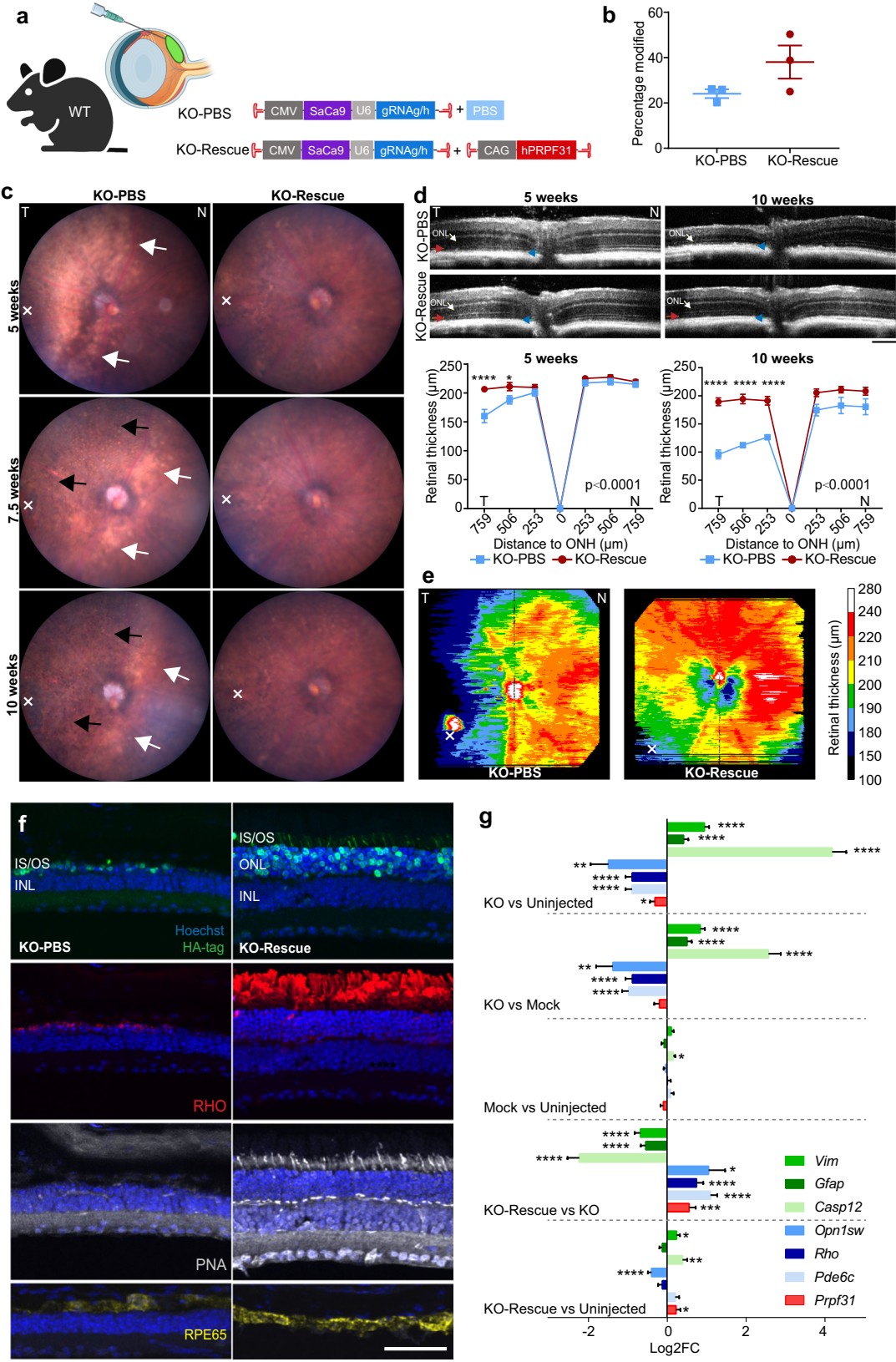

dissected from the center, middle, and peripheral retina, then incubated on a transmembrane, in a medium containing AAV particles (see Methods). AAV-CAG-GFP was applied to retinal explants as control for monitoring of the transgene expression during culturing, and CMV-SaCas9-gRNAa/c/d or a/b were applied to KO explants. AAV-7m8 or K9#12 was used as they have been shown to infect primate retinas with high efficiency[38,40,41]. Fresh medium was replaced every second day, and retinal explants were harvested 9 days post AAV infection. Strong GFP expression was observed in the outer retina in cross-sections and flat-mounted retinal explants (Fig. 9a, b). Immunolabeling of anti-HA tag confirmed the successful expression of HA-tagged Cas9 in explants (Fig. 9b). Genomic DNA from the most highly infected areas of retinal

**Fig. 7 | Gene augmentation preserves the structure of *Prpf31* knockout retinas.**
**a** Illustration of experimental approach. KO-Rescue eyes were subretinally injected with KO vector 7m8-CMV-SaCas9-U6-gRNAg/h plus rescue vector 7m8-CAG-hPRPF31. KO-PBS eyes were injected with KO vectors plus PBS. **b** Deep sequencing of DNA collected from whole retinas at 10 weeks (p.i.) showed that KO rescue eyes had a higher percent of edited *mPrpf31* than KO-PBS eyes (*n* = 3). **c** Fundus images. Injection sites (x) were located at the temporal retina. KO-PBS eyes showed expanding areas of retina pallor (white arrows) and retinal pigmentation (black arrows) over time. In contrast, no prominent areas of retina pallor were observed in KO-Rescue eyes, and pigmentation was found only in a limited area surrounding the injection site (KO-PBS *n* = 16, KO-Rescue *n* = 16). **d** OCT B-scans of retinas at the optic nerve head (ONH) and quantification of retinal thickness (neuron retina and RPE), *n* = 6. ONH, optic nerve head; T, temporal; N, nasal. Scale bar vertical, 100 μm; Scale bar horizontal, 200 μm. **e** Heatmaps of retinal thickness at 10 weeks (p.i.) KO-PBS eyes showed significant retinal thinning at the temporal retina, while

KO-Rescue eyes had improved thickness across the retina with damage limited to the injection site (x). **f** Immunolabeling of HA-tagged Cas9 (green), and specific markers for rods (RHO in red), cones (PNA in grey) and RPE (RPE65 in yellow). KO-Rescue eyes have maintained retinal structure including ONL, rods, cone, and RPE, despite expression of KO vectors. Scale bar, 50 μm. **g** Differential gene expression at 10 weeks (p.i.) (*n* = 3). KO-Rescue eyes showed increased total *PRPF31* expression, downregulated cell death and glial cell activation marker gene, and upregulated rod and cone marker genes relative to KO eyes. Data are shown as mean ± SEM. For retinal thickness, the significance of the overall treatment factor was calculated by two-way ANOVA and shown as a *P* value in each plot. The significance of the treatment factor at each distance from the ONH was calculated by Sidak multiple comparisons test and marked as *$P$ < .05; **$P$ < .01; ***$P$ < .001; ****$P$ < .0001. ONH = optic nerve head. IS/OS = inner segments/outer segments; ONL = outer nuclear layer; INL = inner nuclear layer. Source data are provided as a Source Data file.

explants were extracted and genomic regions containing gRNA targets were PCR amplified. The 7m8-SaCas9-gRNAa/b infected human explants showed a 12.76 ± 1.43% modification of *PRPF31* quantified by deep sequencing, resulted in a 28 bp deletion between gRNAa and gRNAb (Fig. 9c). K9#12-SaCas9-gRNAa/c/d infected human and rhesus retinal explants and resulted in truncated DNA (Fig. 9d–f), with the size of the deletion corresponding to length of sequence between gRNAa and gRNAc (Supplementary Fig. 1c), suggesting that gRNAa and gRNAc were more efficient than gRNAd. The truncation of DNA between gRNAa and gRNAc was also verified by Sanger sequencing (Supplementary Fig. 10). Western blotting confirmed the expression of HA-tagged Cas9 protein in KO samples, and revealed the substantial reduction of PRPF31 protein in K9#12-SaCas9-gRNAa/c/d treated human retinal explants compared to control retina explants (Fig. 9g, h). These experiments strongly support the feasibility of developing NHP models of PRPF31-RP using the Cas9 vectors developed here.

## Discussion

In this study, we used AAV-mediated CRISPR/Cas9 KO constructs delivered via subretinal, intravitreal, or systemic injection to create mouse models of PRPF31-RP, resulted in varying levels and rates of retinal dysfunction. Subretinal delivery of *Prpf31* KO vectors resulted in obvious retinal pigmentation, severe retinal structural changes, and functional disruption within 10 weeks (p.i.). This is the first mouse model with a comparable phenotype to human PRPF31-RP patients, providing a new platform to investigate the pathogenetic mechanisms of PRPF31 and to evaluate treatment efficacy in vivo.

CRISPR/Cas9 edits a gene of interest through creation of a double-strand break in genomic DNA at the gRNA-targeted sequence. AAV-CRISPR/Cas9-based KO of *Prpf31* in vivo may result in a mixture of cells with mutation in both alleles, mutation in a single allele, and non-edited wildtype cells. Mediated by AAV delivery, constitutive expression of Cas9 and/or gRNA in AAV-infected cells, over time, likely results in KO of both alleles. In contrast, in humans, PRPF31 mutations are present from the embryonic stage and cause disease through haploinsufficiency[1,27]. The models developed here progress faster and have more severe retinopathy than what is observed in human patients, likely due to the complete KO of *Prpf31* in AAV-infected retinal cells. However, *Prpf31*-KO in the mouse retina closely models the major structural and functional changes observed clinically in PRPF31-RP. Subretinal delivery of *Prpf31* KO vectors resulted in retinal pigmentation and attenuated retinal vessels. The ONL was severely thinned, and ERG recordings revealed a reduction in a- and b-wave amplitudes over time. Alterations in the RPE and retinal glia activation were also observed.

Intravitreal delivery of *Prpf31* KO vectors to the mouse retina using 7m8, an AAV2-based vector developed through directed evolution for its ability to deliver genes to the outer retina in mice, resulted mainly in changes in the inner retina and lesser changes in the outer

retina. This corresponds to the transfection pattern of 7m8, which expresses most strongly in inner retinal cells from the vitreous, when packaged with a ubiquitous promoter[38]. While outer retinal thinning is well-described in PRPF31-RP patients, there is little clinical data available on the condition of inner retina. Changes in inner retina in RP patients have largely been understood as a result of photoreceptor degeneration[42]. Histological analysis of human and mouse retina localize the expression of PRPF31 in inner retina as well as outer retina[43–45]. Our study showed that inner retina including ganglion cells and bipolar cells were affected as a primary result of *Prpf31* KO. It may be of interest to examine the inner retina when evaluating disease progression in PRPF31-RP patients, in order to determine the most effective gene therapy approach, and while considering new approaches for vision restoration such as optogenetic therapies, which hold great promise for late-stage RP patients but require RGCs to be in good condition[46].

The retina is a highly biosynthetically active tissue, and previous studies have suggested that the retina has a relatively higher demand for the functions of PRPF31, resulting in increased sensitivity to the reduction of PRPF31[35,47,48]. This may explain why PRPF31 mutant carriers develop disease specific to the retina, as the remaining PRPF31 levels provided by the WT allele are sufficient for the normal function of other organs and tissues. Germline homozygous KO of *Prpf31* in mice leads to abnormal embryonic development, while heterozygous KO resulted in no developmental abnormality[25,26]. We performed systemic KO of *Prpf31* in neonatal mice, resulting in stunted development and early death within 4 weeks (p.i.). This indicates that PRPF31 plays critical roles ubiquitously and that it is required for postnatal survival as well as embryonic development. Systemically delivered *Prpf31* KO vectors in P0 mice may provide an animal model for studying the biological function of PRPF31 across tissues.

The gene augmentation rescue studies performed here were carried out using co-injection of KO and therapeutic vectors, in order to allow the earliest rescue possible, since retinal degeneration in this model progresses rapidly. In humans, the disease-causing mutation is present before birth and often diagnosed in adolescence. Further studies using sequential delivery of KO and rescue vectors on a more slowly degenerating background will be required to explore the potential of gene augmentation to reverse retinal degeneration with timing more similar to the course of human disease. To our knowledge this is the first use of in vivo CRISPR-KO and augmentation to simultaneously create and treat a model of retinal dystrophy. This approach may be useful for the rapid animal model creation of other homozygous lethal conditions. The expression of *hPRPF31* was driven by a ubiquitous CAG promotor, as our study showed that a wide variety of retinal cells were affected by *Prpf31* KO. The disease-causing cell type(s) of PRPF31-RP are most widely thought to be photoreceptors and/or RPE cells[25,32,33,49]. Studies using cell type-specific promoters to drive KO and/or rescue may provide further insight into the role of

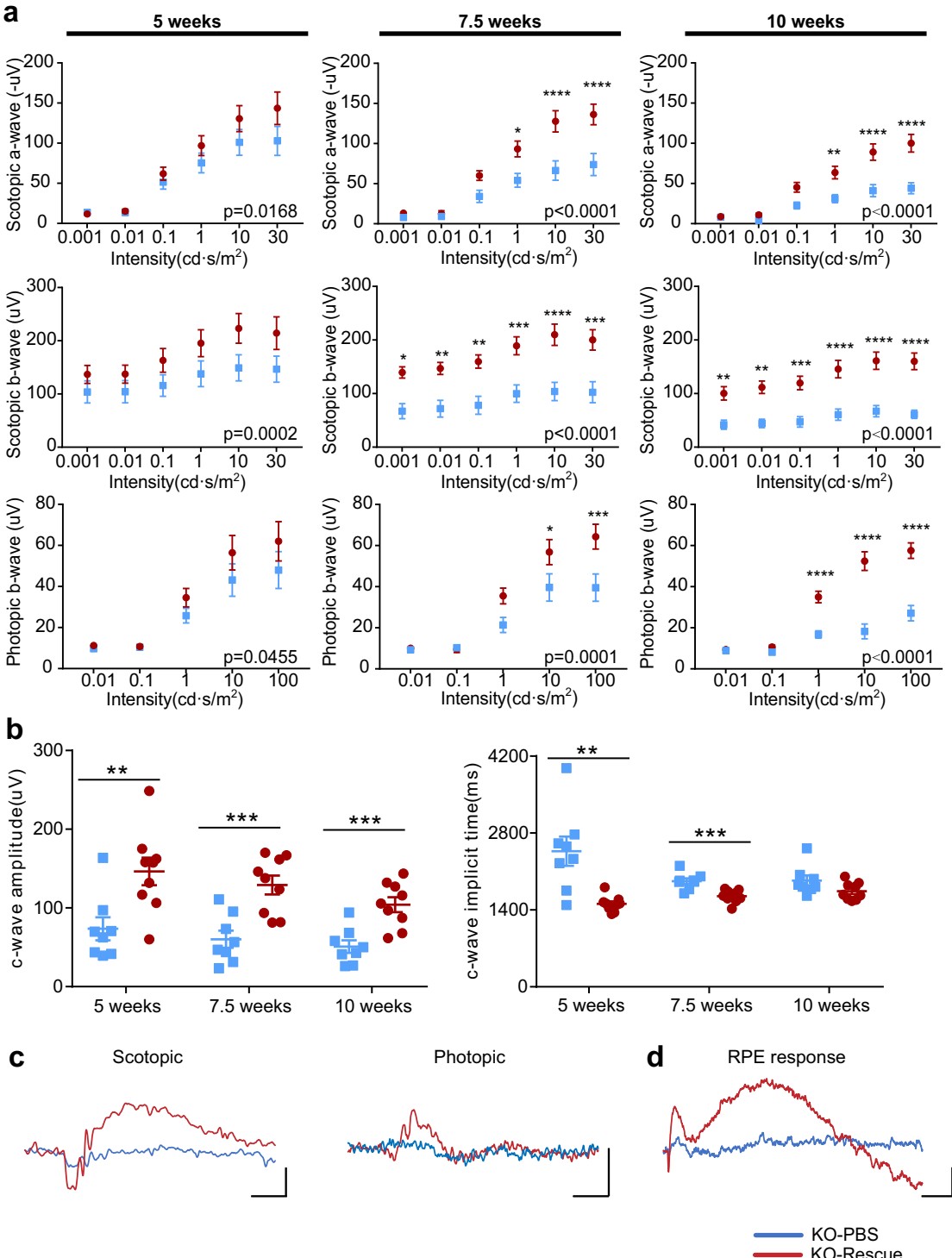

**Fig. 8 | Gene augmentation preserves visual function of the *Prpf31* KO retina.**
**a** Quantification of ERG scotopic and photopic a- and b-waves amplitudes ($n = 8$).
**b** ERG recordings of dark-adapted c-waves ($n = 8$). **c** Representative scotopic
(10 cd·s/m² flash) and photopic (10 cd·s/m² flash) ERG recordings 10 weeks (p.i.).
Scale bar vertical, 50 uV; Scale bar horizontal, 50 ms. **d** Representative c-wave
recordings 10 weeks (p.i.) Scale bar vertical, 50 uV; Scale bar horizontal, 500 ms.
KO-PBS data points and traces are shown in blue; KO-Rescue is shown in red.

Data were shown as mean ± SEM. The significance of the overall treatment factor
was calculated by two-way ANOVA and shown as a *P* value in each plot. Significance
of the treatment factor at each flash intensity was calculated by Sidak multiple
comparisons test and marked as *$P < .05$; **$P < .01$; ***$P < .001$; ****$P < .0001$. Data of
ERG c-waves were analyzed using a two-tailed unpaired *t*-test. Source data are
provided as a Source Data file.

*Prpf31* in individual cell types. In addition, the development of a tissue-specific inducible Cre-Lox mouse line may be a useful approach to achieving more consistent and controlled ablation to reduce variability caused by retinal injections.

AAV-mediated *hPRPF31* gene augmentation prevented the formation of retinal pigmentation, preserved normal retinal structure and function, including photoreceptors and RPE, and downregulated the expression of cell death and gliosis markers in

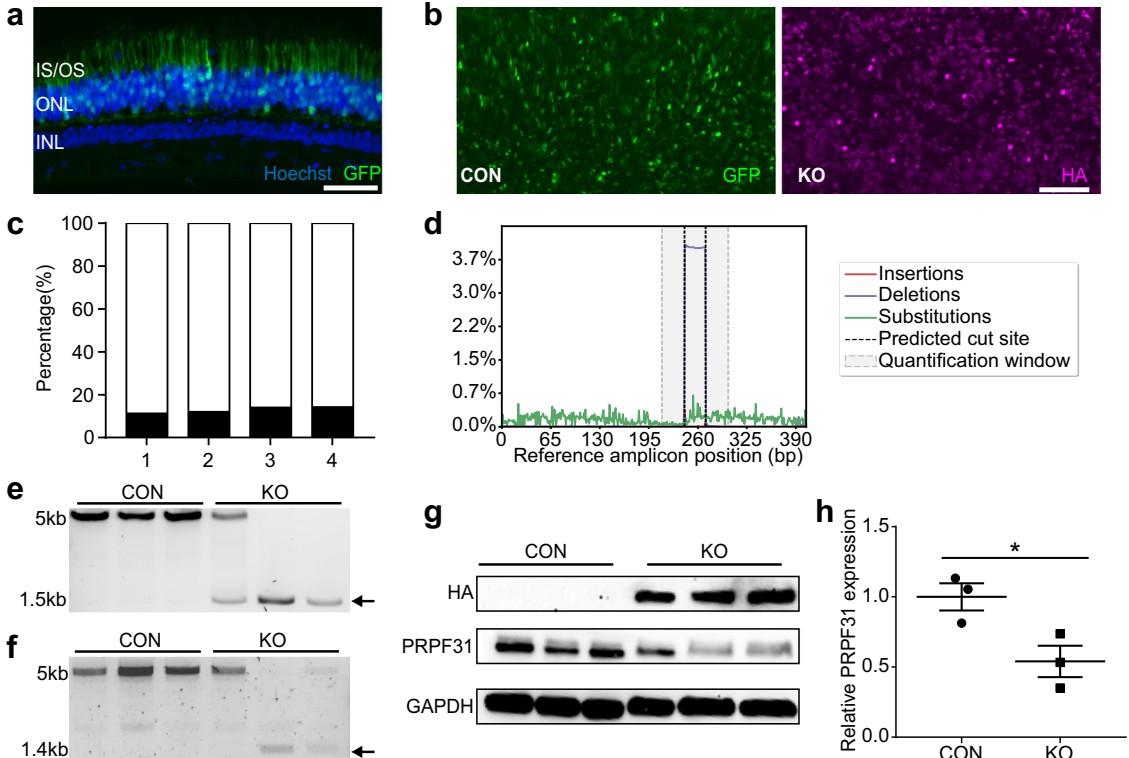

**Fig. 9 | KO of PRPF31 in NHP and human retinal explants demonstrates the feasibility of establishing a non-human primate PRPF31 RP model. a** GFP expression in the outer retina of macaque retinal explant 9 days post AAV infection. Scale bar, 50 μm. **b** GFP expression (left) and immunolabeling of anti-HA tag (right) confirmed the successful expression of HA-tagged Cas9 in human retinal explant flatmounts. **c** Deep sequencing of DNA from the 7m8-SaCas9-gRNAa/b infected human explants revealed editing efficiency of 12.76 ± 1.43% at the gRNAa/b target region of *PRPF31* (*n* = 4). **d** Mutation pattern of modified *PRPF31* from 7m8-SaCas9-gRNAa/b infected human retina explants. The main editing event identified was a -28 bp deletion between the predicted cut sites of gRNAa and gRNAb. **e, f** K9#12-SaCas9-gRNAa/c/d infected human and rhesus(f) retinal explants, resulting in truncated DNA, with the size of deletion corresponding to the distance between gRNAa and gRNAc predicted cut sites (arrows). **g, h** Western blotting revealed that K9#12-SaCas9-gRNAa/c/d-treated human retina resulted in a substantial reduction of PRPF31 protein compared to control retinal explants (*n* = 3). Data are shown as mean ± SEM and were analyzed using an unpaired *t*-test. *P < 0.05. Source data are provided as a Source Data file.

the *mPrpf31* KO retinas. This rescue was achieved in a disease model with more rapid progression and greater severity than patients, further demonstrating the promise of gene augmentation therapy for treating the loss of vision in PRPF31-RP patients.

There are immunological and structural differences between human and mouse eyes, including the presence of a fovea, larger total size, more viscous vitreous and a thicker inner limiting membrane on the retinal surface[50,51]. NHP's are the most biologically and anatomically similar to human, and therefore serve as a critical preclinical animal model. Germline editing may, in the future, be used to produce a PRPF31−/+ primate that more precisely models the haploinsufficiency of PRPF31 patients. However, the creation and maintenance of PRPF31 +/− primates would require years, and would be costly. In comparison, generating PRPF31 mutant NHP models by intraocular injection of AAV-mediated CRISPR/SaCas9 would be relatively achievable and rapidly available, allowing for larger-scale analysis. In this study, we investigated the possibility of creating a PRPF31-RP NHP model by delivering AAV-CRISPR/Cas9 *PRPF31*-KO vectors, similar to the constructs tested in mice, to rhesus and human retinal explants. Our results showed the successful expression of AAV-CRISPR/Cas9 constructs and KO of *PRPF31* in rhesus and human retina, strongly supporting the feasibility of using the *PRPF31*-KO vectors developed here to establish a rapidly available in vivo NHP model of PRPF31-RP. However, retinal ex-vivo cultures do not contain the same structural barriers as in vivo models, and their survival window limited the use for long term studies, thus, an in vivo NHP PRPF31-RP model will be an important objective to pursue.

Collectively, these experiments further advance our understanding of PRPF31-RP and provide important in vivo mouse models, allowing further investigation of PRPF31 pathogenic mechanisms and development of treatments for PRPF31-RP in mice or future NHP models. This work also demonstrates in vivo proof-of-concept for AAV-PRPF31 gene augmentation for the first time. Together, these studies will accelerate therapeutic gene augmentation for PRPF31 to the clinic.

## Methods
### Animals
All procedures were performed in accordance with the Association for Research in Vision and Ophthalmology Statement for the Use of Animals in Ophthalmic and Vision Research. All animal experiments were approved by the University of Pittsburgh Institutional Animal Care and Use Committee (IACUC).

### Human tissue
All experiments were performed with approval and oversight from the University of Pittsburgh Committee for Oversight of Research and Clinical Training Involving Decedents (CORID, approval #927). Eyes from donors was obtained from the Center for Organ Recovery & Education (CORE). Organ recovery was performed by CORE and consent was obtained by CORE. Donor #1, 45-year-old male; donor #2, 32-year-old male, no history of retinal disease.

## gRNA design and construction of AAV vectors

SpCas9 and SaCas9 were used in this study. gRNAs were designed to target early exons of *Prpf31* in mouse, human, and rhesus macaque genome. For SpCas9, gRNA sequences were designed using the Synthego Knockout Guide Designer (https://design.synthego.com/#/) based on human and mouse genome, and then two high-ranking gRNA sequences (gRNA1 and gRNA2) that aligned to *PRPF31* coding sequences of all three species were manually selected. The U6-gRNA1-gRNAscaffold and U6-gRNA2-gRNAscaffold fragments were synthesized (Integrated DNA Technologies) then cloned into pAAV-CAG-GFP vector[38] to create the pAAV-U6-gRNA-CAG-GFP construct.

gRNAs for SaCas9 were designed using CHOPCHOP[52] and CRISPOR[53]. gRNAa, gRNAb, gRNAc, gRNAd were designed to target human and rhesus macaque *PRPF31* coding exons. gRNAg, gRNAh, and gRNAk were designed to target mouse *Prpf31* coding exons. gRNA sequences were cloned into pX601-AAV-CMV_NLS-SaCas9-NLS-3xHA-bGHpA_U6_BsaI-sgRNA (Addgene, plasmid #61591). The unaltered Addgene plasmid #61591 with non-targeting gRNA to m*Prpf31* was used as the mock Cas9 vector. All gRNA sequences are provided in Supplementary Table 2.

To construct the rescue vector pAAV-CAG-hPRPF31, the *hPRPF31* coding sequence was obtained from NCBI (https://www.ncbi.nlm.nih.gov/nuccore/BC117389.1) and gene fragments were synthesized (Integrated DNA Technologies) and cloned into the pAAV-CAG backbone from the pAAV-CAG-GFP plasmid.

## gRNA screening

For testing the efficiency of SpCas9 gRNA, HEK293 cells were co-transfected with the pX551-CMV-SpCas9 (Addgene, plasmid#107024) and pAAV-U6-gRNA1-CAG-GFP or pAAV-U6-gRNA2-CAG-GFP plasmid at a ratio of 1:1. Cells were harvested 72 h post-transfection and GFP positive cells were sorted using a FACSAria II Flow Cytometer (BD Bioscience). Genomic DNA from GFP-positive cells was extracted (Qiagen, DNeasy Blood & Tissue Kits), and the surrounding region of the gRNA target sites was PCR amplified and Sanger sequenced. Editing efficiency was analyzed by the ICE V1 CRISPR Analysis Tool (https://ice.synthego.com/#/). All primer sequences are provided in Supplemental Table 3.

The efficiency of SaCas9 gRNAs was tested by co-transfecting HEK293 cells (for human/ rhesus macaque gRNAs) and NIH/3T3 cells (for mouse gRNAs). Cells were harvested 72 h after transfection, and the expression levels of *PRPF31* mRNA were quantified by qRT-PCR. All primer sequences are provided in Supplemental Table 4.

Of the two SpCas9 gRNAs tested, gRNA2, which targets mouse, human, and rhesus macaque *PRPF31* exon 6, was the most efficient (Supplementary Fig. 1a, b, and methods). Co-transfection of SpCas9 and gRNA2-GFP plasmids resulted in $92.3 \pm 0.58\%$ editing of *PRPF31* in GFP positive HEK293 cells.

SaCas9 is smaller than SpCas9, allowing for packaging of the Cas9 protein and gRNA into a single AAV vector. Three SaCas9 gRNAs that target mouse *Prpf31* (m*Prpf31*) were individually tested. To achieve higher knock-out (KO) efficiency, multiple gRNA plasmids were transfected simultaneously. Co-transfection of SaCas9-gRNAg/h plasmids outperformed all other gRNA combinations and resulted in the lowest m*Prpf31* expression level ($39.00 \pm 16.09\%$ relative to an untransfected control) in NIH/3T3 cells. gRNAg targets m*Prpf31* exon 2 and gRNAh targets exon 3.

There is no available SaCas9 gRNA that targets both mouse and human/ rhesus macaque PRPF31, due to cross-species sequence differences. For gRNAs targeting human/ rhesus macaque *PRPF31*, four gRNAs were tested: gRNAa (exon 2), gRNAb (exon 2), gRNAc (exon 4), and gRNAd (exon 5). To achieve higher knock-out (KO) efficiency, multiple gRNAs were used simultaneously. Co-transfection of SaCas9-gRNAa/c ($44.33 \pm 6.66\%$ *PRPF31* expression relative to control) and SaCas9-gRNAa/d ($51.67 \pm 7.57\%$ *PRPF31* expression relative to control)

showed the highest knockout efficiency among the combinations tested. The combination of SaCas9-gRNAa/b also resulted in considerably reduced *PRPF31* expression ($56.67 \pm 9.03\%$ *PRPF31* expression relative to control).

## Cell culture and transfection

HEK293 cells (Cell Biolabs AAV293 cells # AAV-100) and NIH/3T3 cells (ATCC, #CRL-1658) were maintained in DMEM + Glutamax (Gibco) supplemented with 10% fetal bovine serum, 37 °C, 5% CO2. Transfection of HEK293 cells was performed according to the manufacturer's protocol of DNA transfection reagent (JetPrime, #114-15) with 800 ng plasmid per well of a 12-well plate. Transfection of NIH/3T3 cells was performed with Lipofectamine 3000 (Thermo Fisher Scientific) with 1.5 µg plasmid per well of a 12-well plate. The medium was replaced 4–6 h after transfection. Cells were harvested 72 h post transfection.

## AAV production

All constructs for AAV packaging were sequenced prior to packaging with Sanger sequencing. Inverted terminal repeats (ITRs) were validated through SmaI digest or deep sequencing (Azenta). AAV vectors were produced in AAV293 cells (Cell Biolabs # AAV-100) using a triple transfection method[54]. Recombinant AAVs were purified by iodixanol gradient ultracentrifugation, buffer exchanged and concentrated with Amicon Ultra-15 Centrifugal Filter Units (#UFC8100) in PBS and were titered by quantitative PCR relative to a standard curve using ITR-binding primers[55].

## Mice

C57BL/6 J mice and Cas9-expressing mice (H11^Cas9 CRISPR/Cas9 knock-in mice, B6J.129(Cg)-Igs2 < tm1.1(CAG-cas9*)Mmw >/J, Stock# 028239) were purchased from Jackson Laboratories. Mice were bred and housed in the animal facility of University of Pittsburgh under standard 12-h light/12-h dark conditions in rooms at a temperature of 65–75 °F (-18–23 °C) with 40–60% humidity.

## Rhesus macaque

ID: M333-16, a 5-year-old female, was housed in the animal facility of the University of Pittsburgh under standard 12 h light/12-h dark conditions. The rhesus was euthanized for other experimental purposes and the eyes were a gift from William Stauffer's Lab.

## Intravenous injections

P0 neonatal pups were anesthetized on ice for 30–60 s, and transferred to an operating microscope[56]. A total of 50 ul volume of AAV or PBS was injected through the temporal vein by a 31-gauge insulin syringe (Becton, Dickinson and Company). A successful injection was confirmed through observation that there was no obstruction from the needle, no bump under the pup's skin and blanching of the vein. Following the injection, pups were warmed, recovered on a heating pad, and confirmed to be in good condition before being returned to their cages. Male and female pups were evenly distributed in each treatment group.

## Intravitreal and subretinal injections

Mice age 4–5 weeks were anaesthetized by intraperitoneal injection of ketamine (80 mg/kg) and xylazine (8 mg/kg) and their pupils were dilated with 1% tropicamide (NDC 17478-102-12) followed by 2.5% phenylephrine (NDC 17478-201-02). Mice were placed under an operating microscope on a heating pad. To visualize the fundus, the mouse eye was covered by a 12 mm round cover slide (Fisher Scientific, 12CIR-2) after applying GenTeal lubricant eye gel (Alcon Laboratories). A guide hole was made with a 30-gauge needle (Medline) through the sclera behind the iris at a 45° angle towards the optic nerve. A micro-injection syringe with 33-gauge blunt-end needle (Hamilton) was then inserted into the hole and AAV was delivered to the vitreous body (for

intravitreal injections), or the needle was used to pierce the temporal retina and inject AAV to the subretinal space (for subretinal injections) over a time course of 20 s. The syringe was held still and remained in the vitreous body for 20 s before slowly being removed from the mouse eye. Left eyes and right eyes were injected with different treatment methods to allow for internal controls. For each treatment group, left and right eyes were evenly distributed. Mice were allowed to recover on the heating pad before fully awakening, and then were returned to their cages. Both male and female mice were used in all experiments.

## Fundoscopy

Eyes were imaged using a Phoenix Micron IV fundus camera system. Anesthesia and pupil dilation methods were the same as described above.

## Optical coherence tomography (OCT)

OCT was performed using ultrahigh-resolution spectral domain optical coherence tomography (SD-OCT) (Bioptigen). Mice were anesthetized, pupils were dilated, and mice were positioned allowing for the optic nerve head (ONH) to appear in the center of the image. B-scans (5 frames averaged) and full field volume scans (300 frames) were captured. The thickness of retinal layers was measured from B-scan images using Image J, and the retinal thickness heatmaps were plotted from volume scans with Bioptigen InVivoVue 3.0.8 software.

## Electroretinogram (ERG)

To evaluate retinal function, ERG recording was performed using a Celeris system (Diagnosys) in a dark room. Mice were dark adapted overnight (>8 h). Following anesthesia, mouse pupils were dilated, and eyes were lubricated by GenTeal lubricant eye gel before placing the electrodes onto the cornea. Mice were kept warm during ERG measurement with a 37 °C heating pad. Scotopic ERG responses to 0.001, 0.01, 0.1, 1, 10, 30 cd·s/m² white flashes were recorded and averaged from three sweeps per flash intensity with intervals of 10 to 30 s between each stimulus. Photopic ERG responses were recorded after a 10 min light-adaptation under 10 cd/m² background light. The intensities of the white flash stimulus were 0.01, 0.1, 1, 10, and 100 cd·s/m² on top of the 10 cd/m² background illumination. The a- and b-wave amplitudes were evaluated from the scotopic and photopic ERG recordings. The dark-adapted c-wave was recorded[49] using a single green flash at 64 cd/m² for 200 ms. Narrow and broad band filters were set to 0.1 and 30 Hz, respectively, and the recording time was 4000 ms. The amplitudes and implicit times of c-waves were evaluated.

## Retinal explant culture

Eyes from rhesus macaque and human donors were dissected within 30 min after surgical removal. Eye cups were flat mounted and vitreous body was removed as much as possible. Central, mid-peripheral and peripheral retina was collected using a 5 mm biopsy punch (Fisher Scientific). Retinal tissue was carefully separated from RPE/choroid/sclera and transferred to cell culture inserts with 0.40 μm pore size (Thermo Fisher Scientific, #140640) with the photoreceptor side attached to the membrane. The retinal explants were cultured in Neurobasal Plus Medium (Gibco, A35829-01) supplemented with 2% B-27 (Gibco, A35828-01), 5 μg/ml Plasmocin prophylactic (InvivoGen, #ant-mpp) and Penicillin-streptomycin (Genesee scientific, PSL01-100ML) at 37 °C, 5% CO₂. The culture medium was changed after 24 h and ~40 μl of AAV suspension (for titers see Supplementary Table 3) was applied to the fresh medium. Medium was completely replaced every 48 h and 40 μl of AAV suspension was added to the fresh medium. GFP expression in the control retina was monitored using an ECHO Revolve flourescence microscope. After an incubation period of 10 days, retinal explants used for histological analysis were fixed with 4% paraformaldehyde (PFA). For those retinas used for DNA and protein extraction, the 1 mm edges of each explant were collected.

## PCR and DNA deep sequencing

To determine editing efficiency in vivo, PCR fragments containing the targeted site of gRNA were amplified from gDNA using primers with partial Illumina adapter sequences (Supplementary Table 4). PCR products were then purified and sequenced with a 300-cycle paired-end run on a MiSeq Illumina sequencer. Sequencing was performed by Genewiz (https://www.genewiz.com/). For plots, Cutadapt2.10 was used to trim adapter sequences and eliminate all low-quality reads below the Phred score cut-off of 20[57], randomly downsampled to an equal number of reads from each biological replicate, and then combined to create a single file. Editing rates were analyzed and visualized using CRISPRESSO (https://crispresso.pinellolab.partners.org/).

## Quantitative real time-polymerase chain reaction (qRT-PCR)

Total RNA was extracted from cells or tissues using an RNeasy mini kit or AllPrep DNA/RNA/Protein Mini Kit (Qiagen). Total RNA was reverse transcribed with the SuperScript™ IV First-Strand Synthesis System (Thermo Fisher Scientific) using random hexamers. qPCR amplification was performed using the FAST SYBR Green Master Mix (Thermo Fisher Scientific, #4385616) following the manufacturer's recommendations. All primer sequences are provided in Supplemental Table 4.

## Differential expression analysis

At 1, 2, 3, and 4 weeks (p.i.), total RNA was extracted from the temporal retina of KO and PBS-injected eyes. At 10 weeks (p.i.), total RNA was extracted from the whole retina of WT uninjected, PBS, Mock, KO, KO-Rescue eyes. Library preparation was performed using the Truseq RNA Library Prep Kit (Illumina) following the manufacturer's instructions. Total RNA input was enriched for mRNA and fragmented. Random primers initiated first strand and second strand cDNA synthesis. Adenylation of 3' ends was followed by adapter ligation and library amplification with indexing. A NextSeq 500 Illumina sequencer with high output 150 cycles was used to obtain 2 × 75 paired-end reads. We performed quality control of raw Fastq files with FastQC software, and Cutadapt2.10 was used to trim adapter sequences and eliminate all low-quality reads below the Phred score cut-off of 20. Salmon 1.1.0 was used to index and quantify transcripts using a mm10 Salmon index, which was produced with salmon index using a partial selective alignment method. Differential gene expression was analyzed using DESeq2 (2.11.40.5).

## Western Blotting (WB)

Total protein from retinal tissue was extracted in RIPA buffer with 1x complete protease inhibitor cocktail (Roche, ref11697498001) or by the AllPrep DNA/RNA/Protein Mini Kit (Qiagen). For Figs. 5f and 8g, western blotting was performed following the method: Protein was quantified using the QuantiPro BCA Assay kit (Sigma-Aldrich) then mixed with sample buffer (Bio-Rad Laboratories) and denatured at 95 °C for 5 min. 20 μg of total protein was loaded and separated by an SDS-PAGE gel. Protein was then transferred to a PVDF membrane and blocked with 5% no-fat milk in TBST (Tris-buffered saline with Tween 20 detergent) for 1 h at room temperature (RT). The primary antibodies were incubated overnight at 4 °C in TBST with 1% no-fat milk. HRP-conjugated secondary antibodies were incubated for 1 h at RT. Antibodies used in this study are provided in Supplementary Table 5. Blots were developed using SuperSignal West Pico PLUS chemiluminescent substrate (Thermo Fisher Scientific). To re-probe membranes, the PVDF membranes were stripped by incubating at RT for 10–15 min in Restore plus western blot stripping buffer (Thermo Fisher Scientific). For Supplementary Fig. 8, PRPF31 expression levels were analyzed using a Jess ProteinSimple Western capillary system (BioTechne) using anti-PRPF31 primary antibody with HRP-conjugated secondary

antibody and using in-capillary protein normalization. Analysis was performed using Compass software (BioTechne). A 12–230 kDa 25-capillary Jess separation kit (SM-W004) was used for protein separation. Samples were prepared following the manufacturer's instructions using the EZ Standard Pack (PS-ST01EZ). Primary detection antibody anti-PRPF31 (Abcam, ab231782) was utilized at a concentration of 1:50. An anti-Rabbit detection module (DM-001) was used for secondary labeling. Protein normalization module DM-PN02 was used to perform sample normalization. Uncropped blots are presented in the Source Data file.

## Immunohistochemistry (IHC)

Mice were euthanized and the superior side of each eye was labeled using a cauterizing pen. Eyes were enucleated and the anterior segment was removed. The eye cups were fixed in 4% paraformaldehyde on ice for 4–6 h and dehydrated sequentially in 5%, 10%, 20%, and 40% sucrose solutions in PBS at RT for at least 30 min each. Eye cups were embedded in a 1:1 mixture of 40% sucrose in PBS: OCT compound (Fisher Scientific) with the superior side facing upwards. 14 μm sections containing ONH were collected on glass slides and stored at −80 °C until immunolabeling was performed. Cryosections were rehydrated with PBS for 10 min and PBS plus 0.1% TritonX-100 (PBST) at RT for 30 min, and slides were blocked with 5% goat serum in PBST for 30 min at RT. Primary antibodies were incubated for 1–2 h at RT. Secondary antibodies were incubated for 1 h at RT. Antibodies used in this study are provided in Supplementary Table 6. Hoechst 33342 (Thermo Fisher Scientific, 1:5000 dilution) was applied for 12 min to stain nuclei. Images of the mid-temporal retina were taken using an Olympus FV1200 confocal microscope.

## Statistical analysis

Two-way analysis of variance (ANOVA) was used to analyze the retinal thicknesses measured from OCT B-scan and the ERG a-, b- wave amplitudes. Two factors, treatment and distance from ONH (for OCT) or flash intensity (for ERG), were taken into consideration. The significance of treatment factor at each distance from ONH (for OCT) or each flash intensity (for ERG) was calculated by Sidak multiple comparisons test. Data from other assays were analyzed by the unpaired two-tailed $t$-test. Statistical analyses were performed using GraphPad Prism 9 (GraphPad Software, La Jolla, CA, USA). Data were presented as means ± SEMs. The criteria for significance were: not significant, $P > .05$; $*P < .05$; $**P < .01$; $***P < .001$; $****P < .0001$.

## Reporting summary

Further information on research design is available in the Nature Portfolio Reporting Summary linked to this article.

## Data availability

Deep sequencing and genome editing quantification data generated in this study have been deposited in Dryad (https://doi.org/10.5061/dryad.bcc2fqzf1). FastQ files are publicly available. Source data are provided with this paper.

## Code availability

All code and tools used for data analysis have been previously published and are freely and publicly available (see Methods.) For SpCas9, gRNA sequences were designed using the Synthego Knockout Guide Designer (https://design.synthego.com/#/). gRNAs for SaCas9 were designed using CHOPCHOP[52] and CRISPOR[53]. For in vivo editing efficiency analysis, sequencing was performed by Genewiz (https://www.genewiz.com/). For plots, Cutadapt2.10 was used to trim adapter sequences[57]. Editing rates were analyzed and visualized using CRISPRESSO[58]. We performed quality control of raw Fastq files with FastQC software Version 0.11.9, and Cutadapt2.10 was used to trim adapter sequences and eliminate low-quality reads. Salmon 1.1.0[59] was

used to index and quantify transcripts. Differential gene expression was analyzed using DESeq2[60] (2.11.40.5). For Jess SimpleWesterns, analysis was performed using Compass software v6.1.0 (BioTechne).

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

## Acknowledgements

We are deeply grateful to the human donors and their families who made human retinal explant studies possible. We thank the Center for Organ Recovery and Education for facilitating the use of human retinal tissue. We appreciate help from the University of Pittsburgh Division of

Laboratory Animal Resources (DLAR) for animal husbandry. The rhesus macaque eyes were a gift from William Stauffer's Lab. We thank Anthony St. Leger and Nancy Zurowski for assistance with FACS sorting; Jeff Gross, Yibo Xi, and Fuyun Bian for guidance with ERG, OCT, WB and IHC; Esin Öztürk, Serhan Turunç, and Sara Jabalameli for advice on experiments and assistance with cell culture and AAV packaging; and Molly Johnson for advice on computational methods. Funding was provided by the NEI/NIH (UG3MH120094 to LB, and R01EY030991 to YC), Research to Prevent Blindness (LB; Career Development Award), Foundation Fighting Blindness (LB; Individual Investigator Award), the UPMC Immune Transplant and Therapy Center (LB), as well as through a scholarship from the China Scholarship Council, and the research fund from The Third Xiangya Hospital and Xiangya School of Medicine (ZX). RNA sequencing was performed at the University of Pittsburgh Genomics Research Core. We acknowledge support from NIH CORE Grant P30 EY08098 to the Department of Ophthalmology, from the Eye and Ear Foundation of Pittsburgh, and from an unrestricted grant from Research to Prevent Blindness. Biorender (www.biorender.com) was used to create figures.

## Author contributions

Z.X. conceived, planned and executed experiments. Analyzed data. Wrote the manuscript. A.V. planned and executed experiments. J.A.S. conceived experiments, contributed materials, wrote the manuscript. Y.C. supervised work. Analyzed data. Wrote the manuscript. L.C.B. conceived, planned and executed experiments. Supervised work. Analyzed data. Wrote the manuscript.

## Competing interests

J.A.S. is a co-founder of Avista Therapeutics and Vegavect. L.C.B. is an inventor on patent applications on AAV capsid variants and a co-founder of Avista Therapeutics and Vegavect. The remaining authors declare no competing interests.
