## [Peer Review File · Nature Communications]

Gene augmentation prevents retinal degeneration in a CRISPR/Cas9-based mouse model of PRPF31 retinitis pigmentosaREVIEWER COMMENTS

Reviewer #1 (Remarks to the Author):

Dr. Byrne and colleagues in this study developed an animal model of one of the more common forms of retinitis pigmentosa (RP) – a form caused by PRPF31 mutations. There was an engineered mouse made several years ago, but its phenotype was extremely mild (and didn't appear until old age) and so the model may not be relevant.

PRPF31 causes autosomal dominant disease but it may do so by haploinsufficiency. Lack of PRPF31 is lethal systemically and so others have not succeeded in generating knockouts. To get around this, the authors delivered a CRISPR/Cas -Prpf31 knockdown vector to the retina. Then, in additional cohorts, they combined the gene knockdown with a gene augmentation rescue strategy. Both the gene editing and the gene augmentation are performed using AAV (mostly AAV7m8 but also an AAV9 tyrosine mutant).

The studies include multiple components: selecting the guide RNAs/Cas system, testing Prpf31 knockdown efficiency in the mouse retina using 3 different delivery methods, evaluating the phenotype both with retinal function (ERGs) and with structural assays (fundus photos, OCT, histopathology) through a 10 week period. Delivery approaches used AAV7m8, a vector developed previously by Byrne and colleagues and one that is known to target a wide variety of retinal cell types. Injections in different cohorts were subretinal, intravitreal and systemic (through the facial vein). The appropriate controls were carried out. Delivery of the CRISPR knockdown system results in significant reduction of prpf31 protein as well as the DNA changes (which involved mostly a 185bp deletion). Delivery to the outer retina, through subretinal injection, resulted in reduced retinal function, retinal degenerative changes and histopathologic changes associated with retinal degeneration. The results were confirmed by injecting the guide mRNA alone into a transgenic CRISPR/Cas mouse model.

There were surprising findings after intravitreal injection of the CRISPR/RNA combination in that inner retina was affected more than outer retina. This information may be useful for those using AAV7m8 in other studies. Systemic delivery to pups at P0 (done at that timepoint presumably since that provided better visibility to the facial vein) resulted in failure to thrive but decreased prpf31 protein in the liver.

The team then co-injected the knockdown vector with a vector containing “hardened” wildtype human PRPF31 to test the possibility of rescue. The ratio was 0.5:0 and 0.5:1 (KO vs augmentation). They found rescue with the gene augmentation vector through the subretinal route of delivery.

Finally, the group looked at the potential for conducting similar sorts of studies in primate retinal explants (one non-human and one human). While those cultures could not be maintained for long (9 days), there appeared to be knockdown and expression. No attempt was made to look at function or histopathology given other limitations.

This is a well-written, logical and thorough research article. The authors have convincingly generated an animal model and have also corrected it with gene augmentation. Thus, this provides proof-of-concept for a gene augmentation approach of intervention. The authors have pointed out that this strategy has its limitations in that one cannot test intervention easily at different stages of disease. They may be able to address that limitation, however, in future studies

I have a few comments:

- 1) Has anyone used this CRISPR/augmentation approach before to make and treat an animal model? If not, it might be worth saying this is "the first."
- 2) This disease model is somewhat aggressive given that there are findings apparent by 10 weeks. The human disease is not quite as aggressive. It is a real advantage to be able to generate the model quickly, but would there be any merit of further testing ratios of KO to augmentation vectors to look at threshold levels of PRPF31 for disease?
- 3) My understanding is that the identity of the disease-causing cell in RP due to PRPF31 mutations is not completely clear. Given your approach, I assume that you believe it is photoreceptors (since subretinal injection and use of AAV7m8 target these cells efficiently). However, could it also be retinal pigmented epithelium? I suggest that you add a sentence or two to discuss how you might approach this question.

Jean Bennett

Reviewer #2 (Remarks to the Author):

Given the importance of PRPF31 and its ubiquitous involvement in splicing machinery, creating retinal disease models with non-lethal knockouts that degenerate in a timely and desirable manner has been particularly difficult. Here, the researchers present a CRISPR-mediated KO of PRPF31 demonstrated throughout multiple layers of the retina and across several injection routes, resulting in both morphological and functional degeneration in a timely manner (~10 weeks). Researchers then demonstrate rescue on both fronts under co-injected gene therapy. This article provides early justification for PRPF31 gene augmentation, which is novel and impactful. Several suggestions for improvement of this article are suggested below:

In this project, the researchers knocked out the gene and then added it in. Adding a section on how the relevance of the KO can be translated to the human condition would greatly strengthen the breadth of discussion in this manuscript.

Please show the protein levels of the KO. These levels may be more severe than patients in which the knockouts tend to affect 50% of protein.

Please also provide the western blots for the CRISPR-treated eyes.

Researchers should also elaborate on the variability range before and after co-injection. Furthermore, given the well-known substantial variation in these retinal injections, it is important to evaluate how the deviation in injections impacts phenotype and the downstream analysis regarding gene therapy efficacy. A potential approach to bypass these shortcomings includes the development of an allele-specific Cre-Lox mouse line inducible under tamoxifen injection, achieving a more consistent and controlled ablation that does not suffer from the variability of retinal injections.

Given the embryonic lethality of homozygous PRPF31 knockouts, heterozygous patients with only one healthy allele likely diverge substantially from the widespread allele-unspecific KO performed here (which is acknowledged by the author in the discussion and attributed to the more severe pathology observed when compared to patients).

As the therapy was assessed with subretinal injection, it is difficult to conclude how the preservation of inner segments may have influenced the success of gene therapy, which

may not be fully relevant to the patient population.

Lastly, co-injection of the knockout and treatment is not feasible when translated to disease progression, diagnosis, and treatment in patients and may likely overestimate the efficacy of gene therapy at clinical trials.

Xi et al.

Gene augmentation prevents retinal degeneration in a CRISPR/Cas9-based mouse model of PRPF31 retinitis pigmentosa

RESPONSE TO REVIEWER COMMENTS

Reviewer #1 (Remarks to the Author):

Dr. Byrne and colleagues in this study developed an animal model of one of the more common forms of retinitis pigmentosa (RP) – a form caused by PRPF31 mutations. There was an engineered mouse made several years ago, but its phenotype was extremely mild (and didn't appear until old age) and so the model may not be relevant.

PRPF31 causes autosomal dominant disease but it may do so by haploinsufficiency. Lack of PRPF31 is lethal systemically and so others have not succeeded in generating knockouts. To get around this, the authors delivered a CRISPR/Cas -Prpf31 knockdown vector to the retina. Then, in additional cohorts, they combined the gene knockdown with a gene augmentation rescue strategy. Both the gene editing and the gene augmentation are performed using AAV (mostly AAV7m8 but also an AAV9 tyrosine mutant).

The studies include multiple components: selecting the guide RNAs/Cas system, testing Prpf31 knockdown efficiency in the mouse retina using 3 different delivery methods, evaluating the phenotype both with retinal function (ERGs) and with structural assays (fundus photos, OCT, histopathology) through a 10 week period. Delivery approaches used AAV7m8, a vector developed previously by Byrne and colleagues and one that is known to target a wide variety of retinal cell types. Injections in different cohorts were subretinal, intravitreal and systemic (through the facial vein). The appropriate controls were carried out. Delivery of the CRISPR knockdown system results in significant reduction of prpf31 protein as well as the DNA changes (which involved mostly a 185bp deletion). Delivery to the outer retina, through subretinal injection, resulted in reduced retinal function, retinal degenerative changes and histopathologic changes associated with retinal degeneration. The results were confirmed by injecting the guide mRNA alone into a transgenic CRISPR/Cas mouse model.

There were surprising findings after intravitreal injection of the CRISPR/RNA combination in that inner retina was affected more than outer retina. This information may be useful for those using AAV7m8 in other studies. Systemic delivery to pups at P0 (done at that timepoint presumably since that provided better visibility to the facial vein) resulted in failure to thrive but decreased prpf31 protein in the liver.

The team then co-injected the knockdown vector with a vector containing "hardened" wildtype human PRPF31 to test the possibility of rescue. The ratio was 0.5:0 and 0.5:1 (KO vs augmentation). They found rescue with the gene augmentation vector through the subretinal route of delivery.

Finally, the group looked at the potential for conducting similar sorts of studies in primate retinal explants (one non-human and one human). While those cultures could not be maintained for long (9 days), there appeared to be knockdown and expression. No attempt was made to look at function or histopathology given other limitations.

This is a well-written, logical and thorough research article. The authors have convincingly

generated an animal model and have also corrected it with gene augmentation. Thus, this provides proof-of-concept for a gene augmentation approach of intervention. The authors have pointed out that this strategy has its limitations in that one cannot test intervention easily at different stages of disease. They may be able to address that limitation, however, in future studies.

We are deeply grateful for Dr. Bennett's positive review of this work, and appreciative for the thoughtful and careful reading of our manuscript.

I have a few comments:

1) Has anyone used this CRISPR/augmentation approach before to make and treat an animal model? If not, it might be worth saying this is "the first."

We thank Dr. Bennett for this suggestion. We performed a literature search and did not find another example of this approach. A knockdown-and-replace approach was used by the Beltran lab as a mutation-independent therapy for Rhodopsin mutations (Cideciyan et al, PNAS 2018), but we did not find examples of knockdown-and-replace to simultaneously create and treat a model. We have added the following text to line 626 in the discussion "To our knowledge this is the first use of in vivo CRISPR-KO and augmentation to simultaneously create and treat a model of retinal dystrophy. This approach may be useful for the rapid animal model creation of other homozygous lethal conditions."

2) This disease model is somewhat aggressive given that there are findings apparent by 10 weeks. The human disease is not quite as aggressive. It is a real advantage to be able to generate the model quickly, but would there be any merit of further testing ratios of KO to augmentation vectors to look at threshold levels of PRPF31 for disease?

Yes, we agree with Dr. Bennett that this is an aggressive model of PRPF31, likely due to a complete loss of PRPF31 in AAV-transfected cells. We also agree that testing additional ratios of KO to augmentation vectors may provide information of the threshold levels of PRPF31 for retinal disease. This is an interesting topic that we are currently investigating in ongoing work for a future manuscript.

3) My understanding is that the identity of the disease-causing cell in RP due to PRPF31 mutations is not completely clear. Given your approach, I assume that you believe it is photoreceptors (since subretinal injection and use of AAV7m8 target these cells efficiently). However, could it also be retinal pigmented epithelium? I suggest that you add a sentence or two to discuss how you might approach this question.

Jean Bennett

We thank Dr. Bennett for this comment, and we fully agree with this point. The disease-causing cell or cells is not yet definitively known. The use of cell-type specific vectors to deliver CRISPR-KO tools may help to answer this question. We have added the following sentence to line 631 in the discussion to further address this comment: "The expression of hPRPF31 was driven by a ubiquitous CAG promotor, as our study showed that a wide variety of retinal cells were affected by Prpf31 KO. The PRPF31 disease-causing cell type(s) are most widely thought to be photoreceptors and/or RPE cells. Studies using cell type-specific promoters to drive KO and/or rescue may provide further insight into the role of Prpf31 in individual cell types."

Reviewer #2 (Remarks to the Author):

Given the importance of PRPF31 and its ubiquitous involvement in splicing machinery, creating retinal disease models with non-lethal knockouts that degenerate in a timely and desirable manner has been particularly difficult. Here, the researchers present a CRISPR-mediated KO of PRPF31 demonstrated throughout multiple layers of the retina and across several injection routes, resulting in both morphological and functional degeneration in a timely manner (~10 weeks). Researchers then demonstrate rescue on both fronts under co-injected gene therapy. This article provides early justification for PRPF31 gene augmentation, which is novel and impactful.

We thank the reviewer for their constructive and careful review of our manuscript, and for their comment about the novelty and impact of the work.

Several suggestions for improvement of this article are suggested below:

In this project, the researchers knocked out the gene and then added it in. Adding a section on how the relevance of the KO can be translated to the human condition would greatly strengthen the breadth of discussion in this manuscript.

We thank the reviewer for this suggestion, and we have added the following text to line 585 in the discussion. “However, Prpf31-KO in the mouse retina closely models the major structural and functional changes observed clinically in PRPF31-RP. Subretinal delivery of Prpf31 KO vectors resulted in retinal pigmentation and attenuated retinal vessels. The ONL was severely thinned, and ERG recordings revealed a reduction in a- and b-wave amplitudes over time. Alterations in the RPE and retinal glia activation were also observed.”

Please show the protein levels of the KO. These levels may be more severe than patients in which the knockouts tend to affect 50% of protein.

Please also provide the western blots for the CRISPR-treated eyes.

We thank the reviewer and agree that these western blots should be added to the manuscript. We have added Supplementary Figure 8, which shows protein levels for Prpf31-KO, and Prpf31-augmented mice. PRPF31 protein levels were reduced in KO eyes and were restored to normal levels in KO-Rescue eyes. Automated quantitative western blotting (Jess system, ProteinSimple) was performed according to manufacturer’s instructions, using the protein normalization module. Samples from individual biological replicates are shown. Protein levels are shown for uninjected WT mice (Uninjected), PBS-injected eyes (1.5 µl PBS), KO vector-injected eyes (KO; 1.5 µl 7m8-CMV-SaCas9-U6-gRNAg/h, 1.01E+12 vg/ml), KO vector+PBS-injected eyes (KO-PBS; 0.75 µl PBS + 0.75 µl 7m8-CMV-SaCas9-U6-gRNAg/h, 1.01E+12 vg/ml), and KO vector+rescue vector-injected eyes (KO-Rescue; 0.75 µl 7m8-CMV-SaCas9-U6-gRNAg/h, 1.01E+12 vg/ml + 0.75 µl 7m8-CAG-hPRPF31, 1.05E+12 vg/ml). Retinas were collected 10 weeks after injection.

However, it is likely, in our opinion, that levels of protein reduction in KO eyes, quantified through Western blot, are underestimated, due to the loss of cells in which Prpf31 has been knocked out. Cells which are efficiently transduced by the AAV and in which

PRPF31 expression is lost over time, and are thus excluded from analysis using this approach. The dramatic retinal thinning observed in KO mice confirms this loss of retinal cells. This likely results in higher apparent levels of Prpf31 in KO eyes.

Researchers should also elaborate on the variability range before and after co-injection. Furthermore, given the well-known substantial variation in these retinal injections, it is important to evaluate how the deviation in injections impacts phenotype and the downstream analysis regarding gene therapy efficacy. A potential approach to bypass these shortcomings includes the development of an allele-specific Cre-Lox mouse line inducible under tamoxifen injection, achieving a more consistent and controlled ablation that does not suffer from the variability of retinal injections.

The reviewer has raised an important point about variability resulting from the injection procedure. While subretinal injections were evaluated via fundus imaging following the injection to ensure approximately equal coverage of the retina, and intravitreal injections were monitored for any significant reflux, we did observe variability in the efficiency of editing between mice (please see Fig 2b, 4b, 6, Supp. Fig. 2, and Supp. Fig. 3b), likely as a result of variation between injections. We agree that the development of an inducible allele-specific Cre-Lox mouse line could be a useful approach to achieving consistent and controlled ablation to reduce variability caused by retinal injections. Our rationale in pursuing this AAV-mediated approach was to create a method that allows for rapid translation into primates and *ex vivo* human retinas, however, we agree that our results here suggest that the creation of a tissue specific Cre-Lox mouse line would be a worthwhile endeavor. We have added the following text to line 634 of the discussion in response to this comment: “In addition, the development of a tissue-specific inducible Cre-Lox mouse line may be a useful approach to achieving more consistent and controlled ablation to reduce variability caused by retinal injections..”

Given the embryonic lethality of homozygous PRPF31 knockouts, heterozygous patients with only one healthy allele likely diverge substantially from the widespread allele-unspecific KO performed here (which is acknowledged by the author in the discussion and attributed to the more severe pathology observed when compared to patients).

As the therapy was assessed with subretinal injection, it is difficult to conclude how the preservation of inner segments may have influenced the success of gene therapy, which may not be fully relevant to the patient population.

We thank the reviewer for this comment. While we did not assess the effect of the preservation of inner segments on rescue in the current study, we are currently exploring the effects of Prpf31-KO and the efficacy of gene augmentation approaches using intravitreal injections in ongoing work for a manuscript in preparation, which may help to answer this question.

Lastly, co-injection of the knockout and treatment is not feasible when translated to disease progression, diagnosis, and treatment in patients and may likely overestimate the efficacy of gene therapy at clinical trials.

We agree with this comment, and we have added the following text in line 621 of the discussion to address this point. “The gene augmentation rescue studies performed here were carried out using co-injection of KO and therapeutic vectors, in order to allow the earliest rescue possible, since retinal degeneration in this model progresses rapidly. In

humans, the disease-causing mutation is present before birth and often diagnosed in adolescence. Further studies using sequential delivery of KO and rescue vectors on a more slowly degenerating background will be required to explore the potential of gene augmentation to reverse retinal degeneration with timing more similar to the course of human disease.”

REVIEWERS' COMMENTS

Reviewer #1 (Remarks to the Author):

The authors have responded appropriately to all of the suggestions in the initial reviews.